# MERIT: Maximum-normalized Element-wise Ratio for Language Model Large-batch Training

## Abstract

Large-batch training has become a cornerstone in accelerating the training of deep neural networks, yet it poses challenges in optimization and generalization. Existing optimizers like AdamW present performance degradation during language models' large-batch training, due to the information bottleneck of attention layers caused by the sharp increase of max attention logit. While the LAMB optimizer partially addresses this issue, some attention layers still experience sharply increased maximum attention logits. The reason is that $l_2$-norm-based trust ratios in LAMB are less effective in directly influencing extreme weight values. Furthermore, the weight-wise trust ratio in LAMB is error-prone due to overlooking relationships of weight values within rows or columns. Building on these observations, we propose a novel optimizer, MERIT, which leverages the max norm to calculate the trust ratio to directly constrain the max attention logit. Moreover, we further construct element-wise trust ratios to provide more robust update scaling by focusing on local weight structures. Extensive experiments of large-batch training across various sizes of GPT-2 models demonstrate the superior performance of MERIT. Notably, during the training of GPT-2 Medium, MERIT enables the use of a 6k batch size without any performance degradation compared to the standard batch size (480). This work highlights the importance of considering the max attention logit and finer granularity trust ratio calculation in large-batch training. It successfully improves the training stability and paves the way for larger batch usage, enabling faster development and iteration on large language models.

## 1 Introduction

The advent of large language models has revolutionized natural language processing, achieving unprecedented performance across a wide range of tasks (Dubey et al., 2024; Touvron et al., 2023; OpenAI, 2024). However, the increasing size and complexity of language models always result in a high time cost for the training. With the growing availability of powerful GPU clusters and specialized hardware accelerators, large-batch training can dramatically reduce the time required to train state-of-the-art models, making it possible to iterate faster and explore more ambitious architectures by processing more data in parallel.

While large-batch training offers the potential for increased parallelism and faster convergence, it also introduces complex optimization dynamics that can impede model performance and stability (Goyal et al., 2018; Keskar et al., 2017; Shallue et al., 2019). Training large language models with large batches typically encounters two main issues. The first problem is that when a fixed number of training tokens is available, using larger batch sizes reduces the total number of training iterations, limiting its ability to learn fine-grained patterns in the data. Additionally, research has shown that training with large batches often leads to models performing poorly on unseen data. When using AdamW optimizer with a large batch size, Figure 1 shows clear performance degradation, requiring additional training tokens to reach comparable generalization levels.

This paper identifies a crucial problem in large-batch training of language models: we observe the sharp increase of max attention logit in attention layers during the training process using AdamW optimizer (Kingma & Ba, 2017; Loshchilov & Hutter, 2019). The inflated max attention logit can

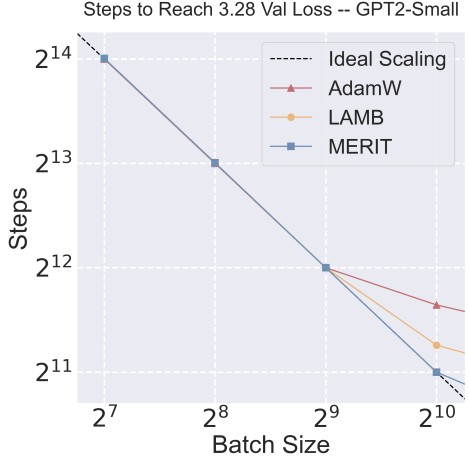 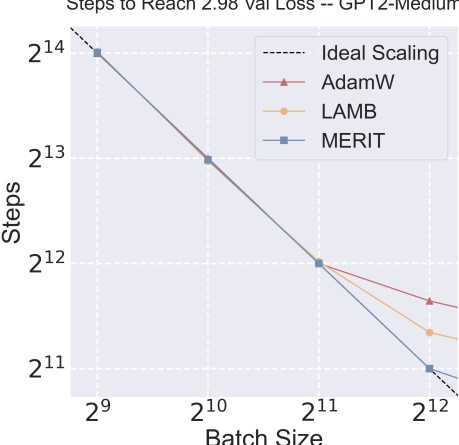

Figure 1: For both GPT-2 models, the connection between batch size and the number of steps required to reach a specific validation loss follows a similar pattern. At first, as the batch size increases, there is a phase of ideal scaling (shown by a dotted line) where doubling the batch size cuts the necessary steps in half. This is followed by a period where the benefits start to decrease. Eventually, a point is reached where further increasing the batch size (data parallelism) offers no additional advantage. This final stage represents the upper limit of large-batch training effectiveness.

result in overly sharp attention distributions, potentially causing the model to focus on specific tokens or patterns overly, thus hindering its ability to capture nuanced relationships in the data (Zhai et al., 2023). While LAMB (You et al., 2020) successfully reduces the max attention logit in the first layer of GPT-2 models (Radford et al., 2019; Brown et al., 2020) by applying a weight-wise trust ratio, it faces limitations in further decreasing the value in the medium layer. The limitation arises from the substantial difference in magnitude between the max norm and the $l_2$ norm of the query/key matrices, which prevents LAMB from directly exerting additional influence on the max attention logit.

Moreover, our analysis reveals that rows and columns in large-batch trained weights often share similarities. The neglect of these relationships in the weight-wise ratio method proposed in LAMB leads to training instability as it fails to mitigate the negative impact of extreme values from other rows or columns. Given rows and columns exhibit high similarity, it allows for calculating element-wise ratios by considering the weights within the same rows/columns, while eliminating influences from other rows/columns. The proposed finer granularity ratios focus on local weight structures, resulting in more stable large-batch training for language models.

Inspired by these insights, we propose a novel optimizer, MERIT, that specifically targets the control of attention logit magnitudes while maintaining the benefits of efficient large-batch training. Our approach builds upon the foundations laid by AdamW and LAMB, introducing max-norm-based trust ratios to precisely modulate the impact of large weights on attention logit distributions and finer granularity ratios for focusing on specific weight structures. We conduct extensive experiments to evaluate the performance of MERIT compared to existing optimizers across various sizes of GPT-2 models. Our findings demonstrate the potential of MERIT to enhance large-batch training by improving convergence properties and generalization performance. This work contributes to the ongoing exploration of optimization strategies in large-batch training and highlights the significance of finer granularity ratio calculation in the design of effective optimizers.

## 2 RELATED WORK

### 2.1 LARGE-BATCH TRAINING

Scaling up batch sizes during the training of deep neural networks has been an active area of research, as it allows for better parallelization across multiple GPUs and reduces time-to-train. How-

ever, naively increasing the batch size often leads to degraded model performance, a phenomenon dubbed the "generalization gap". Several techniques have been proposed to enable large-batch training without compromising accuracy. Goyal et al. (2018) showed that linear scaling of the learning rate with respect to the batch size can maintain model quality for batch sizes up to 8K on ImageNet. Other works proposed novel optimization algorithms like LARS (You et al., 2017) and LAMB (You et al., 2020) that dynamically adapt layer-wise learning rates based on parameter norms and momentum. Liu et al. (2022) introduced a more efficient SAM (Foret et al., 2021) variant for training Vision Transformers (Dosovitskiy et al., 2021) using large batches and Luo et al. (2023) explored memory-efficient optimization techniques for large-batch training of language models. Overall, these methods aim to maintain proper scaling of parameter updates and normalization statistics when scaling up batch sizes. However, LAMB still presents a large max attention logit and shows a weight-wise trust ratio calculation containing some error, leading to large-batch training performance degradation.

## 2.2 MAX ATTENTION LOGIT

The relationship between max attention logit and training stability of transformers has been explored extensively. Researchers have previously documented that Transformer training fails when the attention logits become large. Dehghani et al. (2023) addressed the challenge of uncontrolled growth in attention logits in large-scale transformers by implementing query/key normalization, effectively stabilizing the training process and preventing the near one-hot attention distributions typical in models with parameters nearing 8 billion. Wortsman et al. (2024) observed the loss diverges and training fails when the max attention logit exceeds approximately $10^4$. Zhai et al. (2023) identified attention entropy collapse as a common issue in Transformer training across various domains and tasks and proposed $\sigma$Reparam to reparameterize the weights of linear layers using spectral normalization and a learned scalar. Nevertheless, the significant increase in the maximum attention logit value during large-batch training remains unexplored and leads to the poor performance of existing optimizers.

## 3 PRELIMINARY

### 3.1 MAX ATTENTION LOGIT GROWTH IN LARGE-BATCH TRAINING

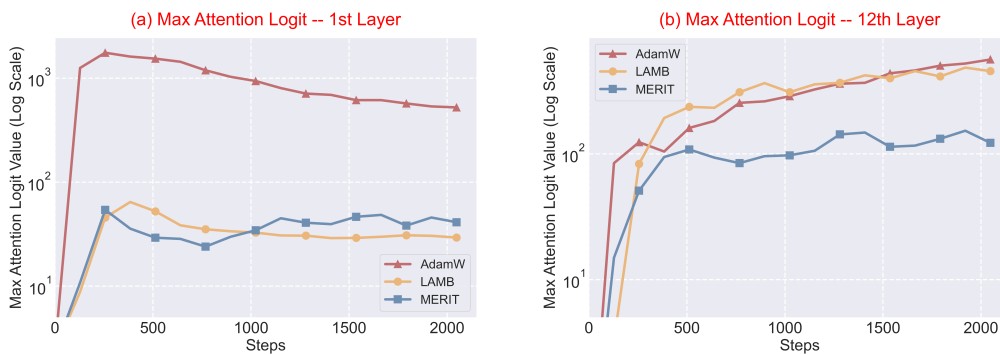

Figure 2: Max attention logit of self-attention layers during the large-batch training of GPT-2 medium model using three optimizers. (a) Max Attention Logit of first self-attention layer. (b) Max Attention Logit of medium (12th) self-attention layer.

In the self-attention layer of a Transformer (Vaswani et al., 2017), attention logits are calculated by combining queries $q_i$ and keys $k_i$ using the formula $z_{ij} = \langle q_i, k_j \rangle / \sqrt{d_h}$, where $d_h$ represents the head dimension. These logits are then processed through a softmax function to generate attention weights, which are subsequently used to aggregate values $v_i$. The maximum attention logit is defined as the max value among the computed attention logits, $\max z_{ij}$. Dehghani et al. (2023) observed that the attention logits $z$ became large when using relatively high learning ratios, which they termed as attention logit growth. Consequently, the attention weights collapse to one-hot vectors and cause unstable training, a phenomenon termed attention entropy collapse by Zhai et al. (2023).

In the large-batch training of GPT models, as the need for larger learning rates than normal batch sizes, AdamW-based training consistently presents a similar max attention logit sharp increase that leads to one-hot-vector attention output that limits the expression ability of attention layers. As shown in Figure 2(a), the max attention logit of the first self-attention layer during large-batch training significantly exceeds the value observed in small-batch training presented in Figure 10, leading to training instability and performance degradation. As a result, AdamW-based large-batching leads to a worse generalization performance compared with small batches.

## 3.2 Trust ratio in LAMB

A distinguishing feature of the LAMB optimizer is its implementation of the "trust ratio", a mechanism designed to dynamically adjust the learning rates for each neural network layer based on their respective weight norms and update norms. The trust ratio $R$ for particular weights $w$ at time $t$ is defined as the quotient of the $l_2$-norm of weights to that of updates $U$:

$$R = \frac{\|w_t\|}{\|U_t + \lambda w_t\|}(U_t + \lambda w_t) \tag{1}$$

where $U_t = \frac{m_t}{\sqrt{v_t} + \epsilon}$ and $\| \cdot \|$ denotes $l_2$-norm. By scaling the learning rates in proportion to these norms, the trust ratio ensures that updates of each layer are neither too large—risking overshooting the minimum—nor too small—leading to slow convergence.

Through the design of trust ratios, LAMB optimizer achieves a balance that allows it to exploit the computational benefits of large batch sizes without compromising the robustness of the model training process. However, the increment of max attention logit still exists during large-batch training as presented in Figure 2(b).

## 4 Algorithm

### 4.1 Maximum Normalized Ratio

The max attention logit is directly relevant to the max norm (largest absolute value) in key matrix $W_K$ and query matrix $W_Q$, as evidenced by the equation: attention logits $a = XW_KW_Q^\top X^\top$, where $X$ represents the input sequence to a self-attention layer. Hence, the issue outlined in Section 3.1 can be addressed by considering extreme values (max norm) when developing the trust ratio for training with large batches. However, Figure 3 demonstrates a substantial disparity between the max norms of query and key weights and their corresponding $l_2$ norms. In this scenario, $l_2$-norm-based trust ratios cannot deal with extreme values of weights effectively. As a result, LAMB often fails to further reduce max attention logit in the medium self-attention layer as depicted by Figure 2(b). Thus, we suggest modifying the LAMB optimizer using the max norm instead of the $l_2$ norm when calculating the trust ra-

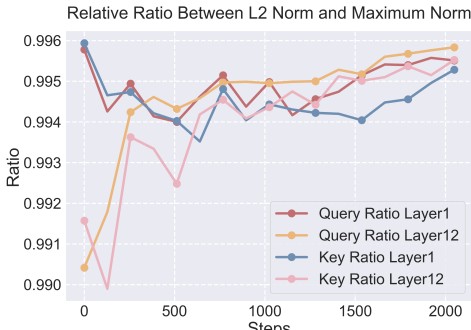

Figure 3: Relative ratios between maximum and L2 norm of self-attention layers in GPT-2 medium. Ratio is calculated as $(\|W\| - \|W\|_m)/\|W\|$, in which $\|\cdot\|$ and $\|\cdot\|_m$ denote $l_2$-norm and max norm.

tio. Therefore, the proposed method gives larger updates to extreme values of weights. This helps prevent extreme values in query and key weights from becoming too large, limiting spikes in the maximum attention logit.

### 4.2 Element-wise Trust Ratio

To further improve the large-batch training performance of layer-wise ratios, we devise an element-wise ratio to capture local weight structures more accurately while maintaining computational efficiency. Due to the multi-headed self-attention mechanism and outlier dimension phenomenon ob-

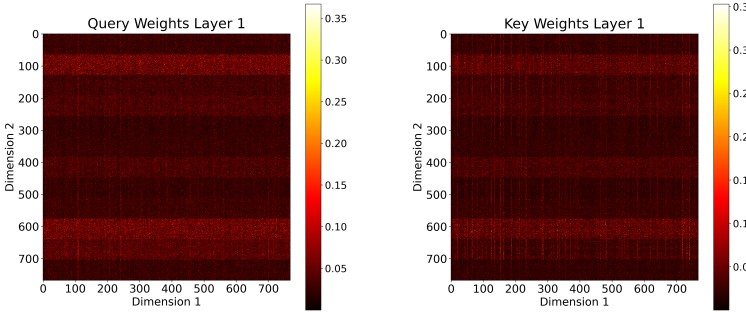

Figure 4: Analysis of GPT-2 small's first attention layer reveals patterns in query-key weight magnitudes: weights show high similarities within both rows (arising from multi-headed attention architecture) and columns (due to the outlier dimension phenomenon).

served throughout the training process of transformers (Kovaleva et al., 2021; Puccetti et al., 2022), weight values exhibit similarities within rows/columns as shown in Figure 4. Given this context, weight-wise trust ratios of LAMB have been observed to introduce certain inaccuracies because extreme values in one row/column can adversely impact the training stability of other rows/columns. To address this limitation, we propose a novel approach that employs an element-wise ratio to leverage the inherent similarity of weights within the same rows or columns of the weight matrix. Our method involves calculating ratios along both rows and columns and then selecting the larger of these two values for each element.

Specifically, let $W \in \mathbb{R}^{n \times n}$ and $U \in \mathbb{R}^{n \times n}$ be the weight matrix and update matrix separately, with $W^{(i)}$ representing elements at the $i$-th row and $W^{(j)}$ representing elements at the $j$-th column. We first calculate the row-wise ratio $R_r$ for each row $R_r^{(i)} = \|W^{(i)}\|_m / \|U^{(i)}\|_m$ and the column-wise ratio $R_c$ for each column $R_c^{(j)} = \|W^{(j)}\|_m / \|U^{(j)}\|_m$. Lastly, the final element-wise ratio $R$ is determined by $R^{(i,j)} = \max\{R_r^{(i)}, R_c^{(j)}\}$. This improved approach seeks to boost the capacity of the optimizer to adjust to specific weight structures in different parts of the network, leading to improved convergence and generalization performance in language models.

### 4.3 MERIT

Algorithm 1 summarizes our proposed MERIT optimizer. The design of MERIT comes from two parts: maximum-normalized trust ratio and element-wise refinement for faster convergence and better generalization performance for large-batch pre-training of language models. Finally, we implement an element-wise clipping mechanism that limits the maximum update magnitude to 1 across all parameter dimensions, which mirrors the update strategy of stochastic Sign Momentum Gradient Descent (Bernstein et al., 2018), serving to enhance the overall stability of the large-batch optimization process. The designs incorporated in MERIT successfully address the issue of rapidly increasing max attention logits in the middle self-attention layers of language models, as illustrated in Figure 2(b).

### 4.4 CONVERGENCE ANALYSIS

**Notation.** Let $\mathbb{I}$ be the $d \times d$ identity matrix, and let $\mathbb{I} = [\mathbb{I}_1, \mathbb{I}_2, ..., \mathbb{I}_h]$ be its decomposition into column sub-matrices $\mathbb{I}_i = d \times d_h$. For $w \in \mathbb{R}^d$, let $w^{(i)}$ be the block of variables corresponding to the columns of $I_i$ i.e., $w^{(i)} = \mathbb{I}_i^T w \in \mathbb{R}^{d_i}$ for $i = 1, 2, \cdots, h$. For any function $f : \mathbb{R}^d \to \mathbb{R}$, $\nabla_i f(w)$ denotes the gradient with respect to $w^{(i)}$. For vectors $u$ and $v \in \mathbb{R}^d$, we use $u^2$ to represent the element-wise square operation and $u/v$ to represent the element-wise division operation. We use $\|\cdot\|, \|\cdot\|_1$ and $\|\cdot\|_m$ to denote $l_2$-norm, $l_1$-norm and max norm of a vector respectively. Consider the following nonconvex stochastic optimization problems of the form

$$\min_{w \in \mathbb{R}^d} f(w) := \mathbb{E}_{x \sim \mathbb{P}}[\ell(w, x)] + \frac{\lambda}{2}\|w\|^2, \tag{2}$$

---

**Algorithm 1** MERIT

---

1: **Input:** $x_1 \in \mathbb{R}^d$, learning rate $\{\eta_t\}_{t=1}^T$, parameters $0 < \beta_1, \beta_2 < 1, \epsilon > 0, m_0 = 0, v_0 = 0$
2: **for** $t = 1$ **to** $T$ **do**
3:      $g_t = \frac{1}{|X_t|} \sum_{x_t \in X_t} \nabla \ell(w_t, x_t)$.
4:      $m_t \longleftarrow \beta_1 m_{t-1} + (1 - \beta_1)g_t$
5:      $v_t \longleftarrow \beta_2 v_{t-1} + (1 - \beta_2)g_t^2$
6:      $u_t = \frac{m_t}{\sqrt{v_t} + \epsilon}$
7:      Weight-wise Ratio $\boldsymbol{b}_t = \frac{\|w_t\|_m}{\|u_t + \lambda w_t\|_m}$
8:      Row-wise Ratio $\boldsymbol{r}_t^{(i)} = \frac{\|w_t^{(i)}\|_m}{\|u_t^{(i)} + \lambda w_t^{(i)}\|_m}$, Column-wise Ratio $\boldsymbol{c}_t^{(j)} = \frac{\|w_t^{(j)}\|_m}{\|u_t^{(j)} + \lambda w_t^{(j)}\|_m}$
9:      Element-wise Ratio $\boldsymbol{s}_t^{(i,j)} = \max\{\max\{\boldsymbol{r}_t^{(i)}, \boldsymbol{c}_t^{(j)}\}, \boldsymbol{b}_t\}$
10:     $w_{t+1} = w_t - \eta_t \cdot \textbf{clip}(\boldsymbol{s}_t \cdot (u_t + \lambda w_t), 1)$
11: **end for**

---

where $w$ is model parameters to optimize, $\ell$ is the loss function and $\mathbb{P}$ is a probability distribution on the unknown training data $\mathcal{X} \subset \mathbb{R}^k$.

**Assumption 1.** The loss function $\ell(u)$ is $L_i$-smooth with respect to $u^{(i)}$, which means there exists a non-negative constant $L_i$ such that

$$|\nabla_i \ell(u, x) - \nabla_i \ell(v, x)| \le L_i |u^{(i)} - v^{(i)}|, \quad \forall u, v \in \mathbb{R}^d, \text{ and } x \in \mathcal{X}, \tag{3}$$

for all $i \in [h]$. Let $L = (L_1, \cdots, L_h)^T$ represent the vector of Lipschitz constants in $h$ dimensions. We use $L_{avg}$ to denote $\sum_i \frac{L_i}{h}$.

**Assumption 2.** The variance in stochastic gradients is subject to the following upper bound:

$$\mathbb{E}|\nabla_i \ell(w, x) - \nabla_i f(w)|^2 \le \sigma_i^2 \text{ for all } w \in \mathbb{R}^d \text{ and } i \in [h]$$
$$\mathbb{E}|[\nabla \ell(w, x)]_i - [\nabla f(w)]_i|^2 \le \tilde{\sigma}_i^2 \text{ for all } w \in \mathbb{R}^d \text{ and } i \in [d], \tag{4}$$

and $\sigma = (\sigma_1, \cdots, \sigma_h)^T$ and $\tilde{\sigma} = (\tilde{\sigma}_1, \cdots, \tilde{\sigma}_d)^T$ are used to denote the vectors of standard deviations of stochastic gradient per layer and per dimension separately.

**Assumption 3.** Gradients are bounded i.e., $[\nabla l(w, x)]_i \le G$ for all $i \in [d]$, $w \in \mathbb{R}^d$ and $x \in \mathcal{X}$. Note that such assumptions are typical in the analysis of stochastic first-order methods.

Due to the usage of element-wise clipping on controlling the worst-case (largest) update size in all parameter dimensions to be at most 1, the training stability is improved and we only need to consider the convergence analysis of weight-wise maximum-normalized ratio that is the lower bound of the proposed MERIT, which is noted as MERIT-W. The following result provides a convergence rate for MERIT-W in general nonconvex settings. Following the analysis in You et al. (2020), we focus on the setting where $\beta_1 = 0$ and $\lambda = 0$.

**Theorem 1.** Let $\eta_t = \eta = \sqrt{\frac{2(f(w_1) - f(w^*))}{\alpha_u^2 \|L\|_1 dT}}$ for all $t \in [T]$, $b = T$, $d_i = d/h$ for all $i \in [h]$, and $\alpha_l \le \|v\|_m \le \alpha_u$ for all $v > 0$ where $\alpha_l, \alpha_u > 0$. Then for $w_t$ optimized by MERIT-W, we have the following bounds:

1. When $\beta_2 = 0$, we have

$$\left( \mathbb{E} \left[ \frac{1}{\sqrt{2 \log(d)}} \|\nabla f(w_a)\|_1 \right] \right)^2 \le O \left( \frac{(f(w_1) - f(w^*))L_{avg}}{T} + \frac{\|\tilde{\sigma}\|_1^2}{Tdh} \right),$$

2. When $\beta_2 > 0$, we have

$$\mathbb{E}[\|\nabla f(w_a)\|^2] \le O \left( \sqrt{\frac{2G^2 \log(d)}{h(1 - \beta_2)}} \times \left[ \sqrt{\frac{2(f(w_1) - f(w^*))\|L\|_1}{T}} + \frac{\|\tilde{\sigma}\|_1}{\sqrt{Td}} \right] \right),$$

where $w^*$ represents an optimal solution to the problem outlined in equation 2 and $w_a$ is an iteration uniformly randomly selected from $\{w_1, \cdots, w_T\}$. For a detailed proof of convergence, please refer to Appendix F.

# 5 EXPERIMENTS

## 5.1 SETUP

**Language modeling.** We conducted large-batch training experiments on OpenWebText (Gokaslan & Cohen, 2019), training autoregressive models from scratch using settings derived from the Chinchilla scaling law (Hoffmann et al., 2022). Following standard protocol, we set the context length of GPT-2 to 1024. Our experiments encompassed three model sizes: 125M (small), 355M (medium), and 770M (large). Detailed specifications of the model configurations can be found in Appendix A.

**Baselines.** We compare MERIT with LAMB, the dominantly used optimizer on large-batch training of language modeling tasks, Adam with decoupled weight decay (AdamW), Lion (Chen et al., 2023), and Sophia-G (Liu et al., 2024). For all models, all learning rates are tuned with grid search. The weight decay is set to 0.1 for all optimizers for a fair comparison. We follow (Liu et al., 2024) for the settings of $\beta$ values: For AdamW: $\beta_1 = 0.9$ and $\beta_2 = 0.95$. For Lion: $\beta_1 = 0.95$ and $\beta_2 = 0.98$. For Sophia-G: $\beta_1 = 0.92$ and $\beta_2 = 0.99$.

**Chinchilla Scaling Law.** The Chinchilla scaling law suggests an optimal training regime of approximately 20 tokens per parameter for large language models. This principle, derived from Hoffmann et al. (2022)'s comprehensive analysis, proposes that model performance is maximized when the number of training tokens scales proportionally with the number of parameters under a fixed compute budget. While the exact ratio may vary slightly depending on specific circumstances, the "20x rule" has been widely adopted as a practical guideline for efficient model scaling in the field of natural language processing (Anil et al., 2023; Muennighoff et al., 2023).

**Implementation.** Following the Chinchilla scaling law, we use batch size 1K for GPT-2 small with 2B training tokens, 4K for GPT-2 medium with 8B tokens, and 8K for GPT-2 large with 16B tokens for the large-batch training setting. Our learning rate (LR) follows a cosine schedule, with the final LR set to 0.1 of the peak LR. We maintain a constant LR warm-up ratio of 0.02 and apply standard gradient clipping (norm) with a threshold of 1.0. In the case of Sophia-G, we select 240 examples from each minibatch to compute the diagonal Gauss-Newton and update the diagonal Hessian every 10 steps. We implement the algorithms in PyTorch (Paszke et al., 2019) and train all the models in bfloat16. All models are trained on H100 GPUs.

**Technical details.** We apply the previously described baseline optimizers and MERIT for three token counts: 2B (Small), 8B (Medium), and 16B (Large). We mainly evaluate GPT-2 models with their log perplexity and plot the validation loss curves. The results from SuperGLUE (Wang et al., 2019), LAMBADA (Paperno et al., 2016), and WikiText (Merity et al., 2017) evaluations are also included in our experiments.

## 5.2 RESULTS

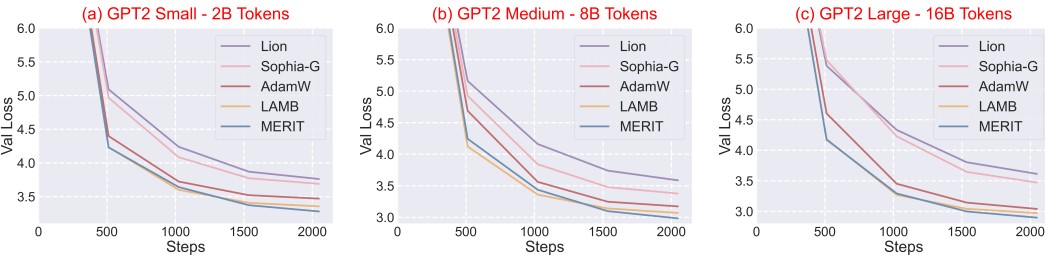

Figure 5: Final validation loss for each optimizer. (a) GPT-2 Small (125M, batch size=1K). AdamW: 3.470, LAMB: 3.355, MERIT: 3.280 (b) GPT-2 Medium (355M, batch size=4K). AdamW: 3.172, LAMB: 3.068, MERIT: 2.982. (c) GPT-2 Large (770M, batch size=8K). AdamW: 3.039, LAMB: 2.971, MERIT: 2.897.

Figure 5 illustrates the validation loss curve on OpenWebText with the same number of steps. MERIT consistently achieves lower validation loss than LAMB, AdamW, Lion, and Sophia-G. As the size of the language model increases, the performance gap between MERIT and baselines becomes larger. Besides, during large-batch training of GPT-2 Large, the performance gap between

AdamW and LAMB diminishes. In contrast, MERIT demonstrates an increased advantage over LAMB under this condition. MERIT achieves a 0.07 smaller validation loss on the 123M model (Figure 5 (a)) with the same training tokens, which means a significant improvement according to training scaling laws in this regime (Kaplan et al., 2020; Hoffmann et al., 2022; Liu et al., 2024).

**The scaling law favors MERIT over LAMB.** Figure 1 illustrates the number of steps required for GPT-2 models with varying batch sizes to reach equivalent validation loss on OpenWebText. The graph reveals a noticeable decline in generalization performance when training language models with large batch sizes using the AdamW optimizer. LAMB's layerwise adaptive optimization strategy mitigates this effect to some extent. Importantly, MERIT enables the use of larger batch sizes without compromising performance (1K for GPT-2 Small and 4K for GPT-2 Medium). Moreover, the performance gap between MERIT and LAMB, given the same number of training tokens, widens for 355M parameter models compared to 125M parameter models.

**Zero-shot Evaluation.** The enhanced validation loss performance translates to better results in evaluation tasks as shown in Figure 6. We measure the zero-evaluation performance of trained GPT models on LAMBADA and WikiText by showing the corresponding perplexity. MERIT successfully obtains lower perplexity across both tasks compared with AdamW and LAMB. Our evaluation focuses solely on zero-shot performance for pre-trained GPT-2 models, as demonstrating in-context learning typically requires GPT models with at least a billion parameters. Additional evaluation results are available in Appendix D.

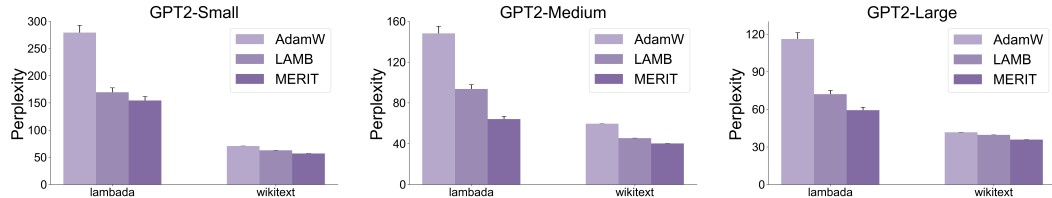

Figure 6: Zero-shot evaluation on LAMBADA and WikiText. With the same number of steps, language models large-batch pre-trained with MERIT outperform models pre-trained with AdamW and LAMB on both tasks with fewer perplexity scores.

## 5.3 FURTHER ANALYSIS

**Comparison of wall-clock time and computational resources.** In Table 1, we present a comparison of the total computational requirements (measured in TFLOPS) per step and the actual time taken (wall-clock time) on A100 GPUs. Following the methodology of Chowdhery et al. (2022), we report the average time per step (T(step)) and corresponding FLOPS. The data in Table 1 reveals that employing maximum-normalized and element-wise trust ratio calculation adds minimal extra computational overhead (1%) compared to $l_2$-norm-based trust ratio of LAMB. Overall, the increase in FLOPS is negligible

Table 1: Wall-clock time and TFLOPS.

| Optimizer | Model Size | T(step) | TFLOPS |
|---|---|---|---|
| AdamW | 770M | 242.50s | 43.91 |
| LAMB | 770M | 243.51s | 43.73 |
| MERIT | 770M | 245.46s | 43.38 |
| AdamW | 355M | 57.69s | 44.05 |
| LAMB | 355M | 57.93s | 43.86 |
| MERIT | 355M | 58.50s | 43.43 |

compared to AdamW and LAMB. Concerning memory usage, our optimizer maintains two memory states, like LAMB and AdamW, resulting in the same memory requirements.

**Performance Gap Between Standard Batch Size and Large Batch Size.** The disparity in generalization between small and large batch sizes during optimization is a complex issue, primarily due to the reduced noise in gradient estimates when using large batches. We conduct experiments following the training protocol outlined in Liu et al. (2024), using 48 billion tokens for training. Our study compares MERIT's performance with large batches against AdamW's performance with small batches (batch size=480). As illustrated in Figure 7 and Table 2, MERIT demonstrates the ability to increase batch sizes to 4K for GPT-2 small pre-training and 6K for GPT-2 medium pre-training without compromising generalization performance. These findings suggest that MERIT enables language models to effectively utilize larger batch sizes as the model scale increases.

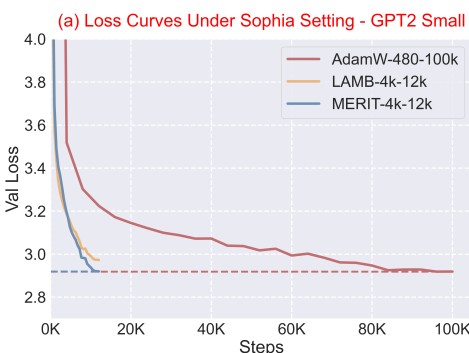 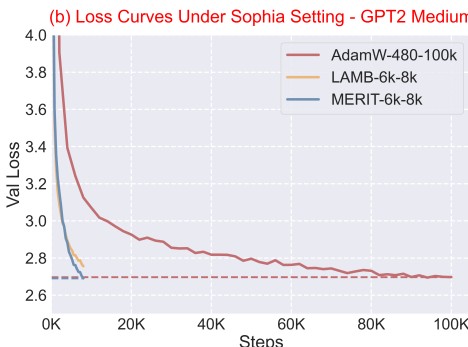

Figure 7: Comparison of validation loss trajectories for GPT-2 Small and GPT-2 Medium models under Sophia training settings.

Table 2: Comparison of zero-evaluation performance for GPT-2 Small and GPT-2 Medium models under Sophia training settings.

| Model | ARC | COPA | HelllaSwag | RACE | WIC | Avg |
|---|---|---|---|---|---|---|
| GPT-2 Small (AdamW-Batch Size=**480**) | 43.43 | 66.00 | 29.20 | 29.00 | 50.16 | 43.56 |
| GPT-2 Small (MERIT-Batch Size=**4k**) | 45.83 | 67.00 | 28.82 | 27.56 | 50.16 | **43.87** |
| GPT-2 Medium (AdamW-Batch Size=**480**) | 49.49 | 71.00 | 32.39 | 30.05 | 50.00 | 46.59 |
| GPT-2 Medium (MERIT-Batch Size=**6k**)) | 50.38 | 70.00 | 32.32 | 30.33 | 50.47 | **46.70** |

**Curvature of Convergence Point.** The curvature of the convergence point in the loss landscape differs significantly between small and large batch sizes, impacting model generalization and robustness. Large batch sizes often lead to convergence in sharper minima with higher curvature Keskar et al. (2017). While these sharp minima may achieve lower training loss, they can result in poorer generalization due to their sensitivity to small changes. In Figure 8(a), we present the eigenvalue distributions of Hessian matrices at the convergence points of GPT-2 small models pre-trained using AdamW and MERIT algorithms. The convergence point achieved by MERIT exhibits a smaller top eigenvalue (12.326) and trace (3444.92) than AdamW whose top eigenvalue and trace equal 37.231 and 12994.91 respectively, and eigenvalues of MERIT are predominantly confined to the range $[-5, 5]$. This reduced spread of eigenvalues suggests that MERIT converges to an overall flatter region in the optimization landscape. Such flat regions, characterized by small eigenvalues and trace, are frequently associated with improved generalization capabilities in language models.

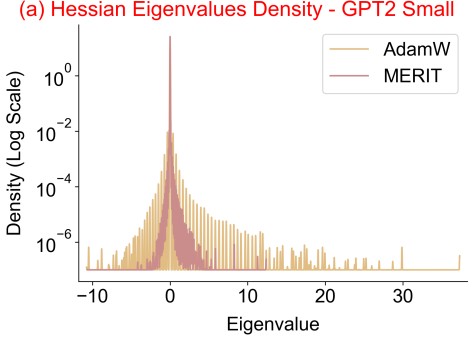 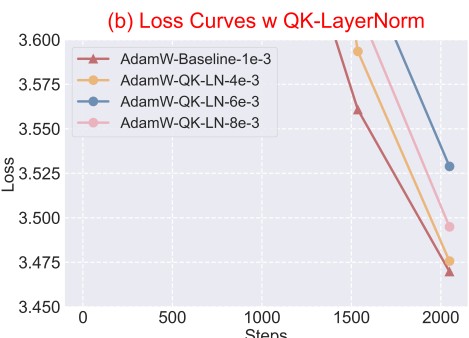

Figure 8: (a) A graphical representation comparing the eigenvalues of Hessian matrices at convergence points, contrasting models pre-trained using AdamW versus MERIT. (b) QK-Norm leads to performance degradation although improving the feasible learning rates of GPT-2 models pre-training without divergence.

**QK-Norm VS MERIT.** QK-Norm (Dehghani et al., 2023) was developed to mitigate training instabilities encountered when scaling Vision Transformer (ViT) models to unprecedented sizes with

higher learning rates. This technique applies Layer Normalization to the query and key vectors prior to the attention computation in the transformer architecture. However, as illustrated in Figure 8(b), while QK-Norm enables larger learning rates for AdamW in GPT-2 models, it negatively leads to performance degradation in large-batch training. A potential explanation for this discrepancy is that QK-Norm aims to stabilize attention computations and inadvertently restricts information flow within the attention layers of language models, which becomes more obvious in large-batch training with much fewer optimization steps. In contrast, our proposed MERIT demonstrates the ability to enhance the performance of language models when trained using large batch sizes.

## 5.4 ABLATION STUDY

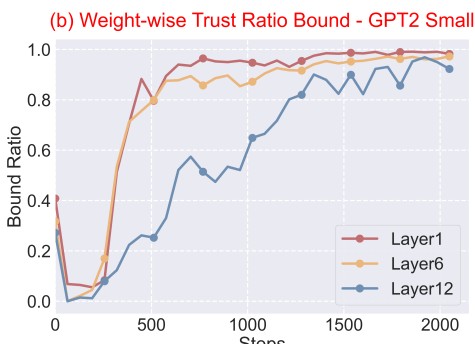

Figure 9: (a) Three ablations present obvious performance degradations. (b) The trigger ratio of the weight-wise trust ratio lower bound during the large-batch training of GPT-2 Medium.

**Ablation 1: Element-wise Clipping.** Figure 9(a) reveals that MERIT, even without element-wise clipping, still outperforms AdamW and LAMB in terms of convergence, albeit with a less pronounced improvement. This finding suggests that when we apply the element-wise trust ratio without clipping, certain elements undergo unexpectedly large update steps, which can adversely affect the language model performance. These results underscore the importance of element-wise update clipping in large-batch training scenarios, where update magnitudes tend to be larger compared to standard training conditions. For visualization of element-wise clipping ratios during GPT-2 Small training, please refer to Appendix E.

**Ablation 2: Weight-wise Ratio Bound.** The implementation of a weight-wise trust ratio as a lower bound for element-wise ratios aims to mitigate excessively small updates during large-batch training of language models. As illustrated in Figure 9(a), the application of this lower bound significantly enhances generalization performance, highlighting the importance of balanced updates across different elements. Figure 9(b) further demonstrates that this lower bound becomes particularly crucial in the latter stages of training. This observation indicates that maintaining a minimum update magnitude grows increasingly important as the model nears convergence. Such a strategy likely enables the model to continue refining its parameters effectively in later training phases, potentially circumventing premature convergence to sub-optimal solutions.

**Ablation 3: Element-wise Ratio.** Element-wise trust ratio calculations enhance the generalization capability of language models by providing more robust ratio estimates focusing on local weight structures for individual weight elements. Figure 9(a) demonstrates the advancement of using element-wise ratio.

## 6 CONCLUSION

Accelerating the pre-training of language models heavily relies on large batch techniques. In this study, we present the MERIT optimizer, which integrates maximum norm and local weight information to compute trust ratios. When applied to the large-batch training of GPT models, MERIT enables larger batch size usage than LAMB and AdamW, while maintaining comparable generalization performance.

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

## A    Models and Hyperparamters Configuration

Table 3: Model Configurations and Peak Learning Rate Under Chinchilla Scaling Law.

| Model | Size | d_model | n_head | depth | Lion lr | Sophia-G lr | AdamW lr | LAMB lr | MERIT lr |
|---|---|---|---|---|---|---|---|---|---|
| Small | 125M | 768 | 12 | 12 | 1e-4 | 1e-4 | 1e-3 | 1e-2 | 9e-3 |
| Medium | 355M | 1024 | 16 | 24 | 8e-5 | 1e-4 | 4e-3 | 1e-2 | 9e-3 |
| Large | 770M | 1280 | 20 | 36 | 8e-5 | 2e-4 | 2e-3 | 8e-3 | 6e-3 |

In our study, we examine three GPT-2 variants: small, medium, and large, as described by (Radford et al., 2019). The specific configurations for these models are outlined in Table A. We utilize the nanoGPT framework (available at https://github.com/karpathy/nanoGPT/) as our codebase. Consistent with nanoGPT's approach, we implement GELU activation functions and omit bias and Dropout (Srivastava et al., 2014) during the pre-training phase.

The GPT-2 models undergo training using the OpenWebText corpus (Gokaslan & Cohen, 2019). We process the text using the GPT-2 tokenizer (Radford et al., 2019). For data organization, we adopt the train-validation split provided by nanoGPT. The training dataset comprises 9 billion tokens, while the validation set contains 4.4 million tokens.

Our training setup employs distributed data-parallel processing with gradient accumulation, allowing for batch sizes of 1K, 4K, and 8K. All model variants are trained using bfloat16 precision. The 125M and 355M parameter models are trained on systems equipped with two H100 GPUs, whereas the 770M parameter models require machines with eight H100 GPUs.

## B    Limitations

**Comprehensive downstream task assessment.** We evaluate large-batch pre-trained models on 7 downstream tasks, which provides valuable but limited insights. A truly comprehensive assessment of language models remains an open research challenge. Our evaluation is further constrained by the modest size of the models studied, which lack advanced capabilities like in-context learning and complex reasoning capability. These limitations indicate the need for caution when extrapolating our findings to larger, more capable models.

**Cross-domain applicability and generalization.** Our study focuses on large language model optimization. However, a truly versatile optimizer should perform well across various domains such as computer vision, reinforcement learning, and multimodal tasks. Due to computational constraints, we have not evaluated the large-batch training performance of our optimizer in these areas. Future work should investigate its efficacy across diverse machine learning paradigms to fully assess its generalizability and potential impact.

**Scaling up to larger language models and datasets.** MERIT has shown promising scalability up to 770M parameter models trained on OpenWebText. While there are no fundamental barriers to scaling further, our comparison with AdamW and LAMB on more extensive models and datasets is constrained by resource limitations. Based on observed improvements in scaling laws and enhanced pre-training stability, we anticipate MERIT to outperform AdamW and LAMB in large-batch training scenarios with larger language models. However, empirical validation of this hypothesis awaits future work with access to greater computational resources.

## C    Max attention logit in small-batch training

Following is the distribution of max attention logits in small-batch (512) training of the GPT-2 medium model using the same chinchilla scaling law setting. Notably, these max attention logits are significantly lower than those observed in large-batch training scenarios. This reduction suggests that the attention outputs are more evenly distributed, which typically leads to improved training convergence and generalization performance.

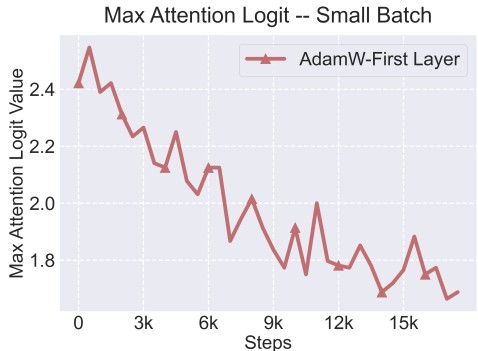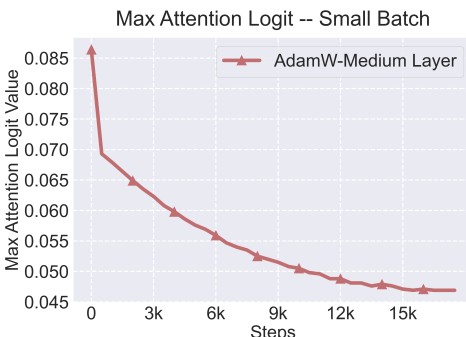

Figure 10: Max attention logit of self-attention layers during the small-batch training of GPT-2 medium model using three optimizers. (a) Max Attention Logit of first self-attention layer. (b) Max Attention Logit of medium self-attention layer.

## D  ZERO-SHOT EVALUATION ON LAMBADA AND WIKITEXT

**Zero-shot Evaluation.** The improved validation loss leads to better downstream task performance, as demonstrated in Figure 11. When comparing models with equal pre-training steps, the GPT-2 variants trained using MERIT consistently outperform those using LAMB and AdamW in zero-shot accuracy across most subtasks.

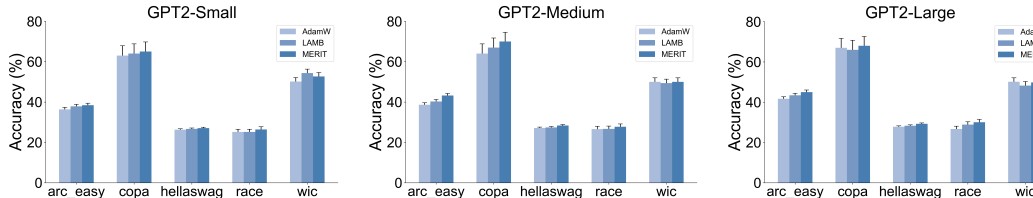

Figure 11: Zero-shot evaluation on SuperGLUE benchmark. Given an equivalent number of training steps, models that undergo large-batch pre-training using MERIT exhibit higher accuracy than those pre-trained with AdamW and LAMB on most tasks.

## E  CLIPPING RATIO IN GPT-2 SMALL LARGE-BATCH TRAINING

Figure 12 presents the element-wise clipping ratio in GPT-2 Small training setting (2B tokens) for the 1st, 6th, and 12th layers.

The analysis of clipping effects across different layers reveals distinct patterns in gradient update behavior during training. The input layer (Layer 1) maintains near-zero clipping ratios throughout, suggesting that early-layer gradient updates rarely require adjustment. In contrast, the middle layer (Layer 6) experiences more substantial clipping, with ratios peaking at 12% during later training stages. The output layer (Layer 12) shows minimal clipping, with ratios reaching only 0.25% at maximum.

This layered pattern demonstrates that the clipping mechanism primarily influences the middle layers, leaving input and output layers unaffected. Such behavior indicates that the clipping mechanism functions as a targeted stabilizer rather than a uniform constraint across the network. Most gradient updates maintain their original direction, with the most significant stabilization occurring in middle layers where feature representations undergo refinement.

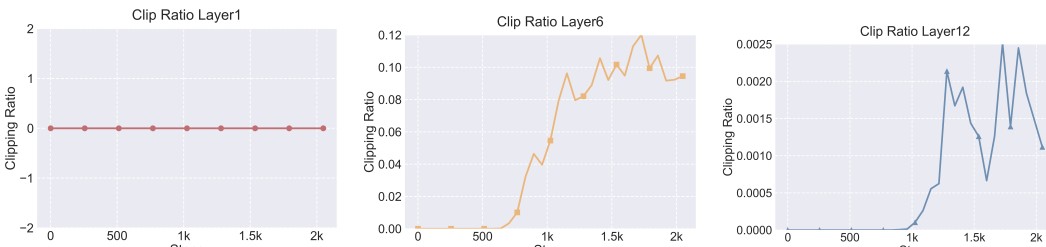

Figure 12: Element-wise clipping ratio for different layers of GPT-2 Small during large-batch training.

## F  CONVERGENCE PROOF OF THEOREM 1

**Proof.** We study how MERIT-W converges across different minibatch sizes. To begin, let's review the update equation of MERIT-W

$$w_{t+1}^{(i)} = w_t^{(i)} - \eta_t \cdot \|w_t^{(i)}\|_m \frac{u_t^{(i)}}{\|u_t^{(i)}\|_m} \tag{5}$$

for all $i \in [h]$.

Since the function $f$ is $L$-smooth, we obtain the following:

$$f(w_{t+1}) \leq f(w_t) + \langle \nabla_i f(w_t), w_{t+1}^{(i)} - w_t^{(i)} \rangle + \sum_{i=1}^{h} \frac{L_i}{2} \|w_{t+1}^{(i)} - w_t^{(i)}\|^2$$

$$\leq f(w_t) \underbrace{- \eta_t \sum_{i=1}^{h} \sum_{j=1}^{d_i} (\|w_t^{(i)}\|_m) \times \left( [\nabla_i f(w_t)]_j \times \frac{u_t^{(i,j)}}{\|u_t^{(i)}\|_m} \right)}_{T_1} + \sum_{i=1}^{h} \frac{L_i d_i \alpha_u^2 \eta_t^2}{2} \tag{6}$$

The above inequality follows from the Lipschitz continuity of the gradient. We bound term $T_1$ in the following manner:

$$T_1 \leq -\eta_t \sum_{i=1}^{h} \sum_{j=1}^{d_i} \|w_t^{(i)}\|_m \times \left( [\nabla_i f(w_t)]_j \times \frac{u_t^{(i,j)}}{\|u_t^{(i)}\|_m} \right)$$

$$\leq -\eta_t \sum_{i=1}^{h} \sum_{j=1}^{d_i} \sqrt{\frac{1 - \beta_2}{G^2 2 \log(d_i)}} \left( \|w_t^{(i)}\|_m \times [\nabla_i f(w_t)]_j \times g_{t,j}^{(i)} \right)$$

$$-\eta_t \sum_{i=1}^{h} \sum_{j=1}^{d_i} \left( \|w_t^{(i)}\|_m \times [\nabla_i f(w_t)]_j \times \frac{u_t^{(i,j)}}{\|u_t^{(i)}\|_m} \right) \mathbf{1}(\text{sign}([\nabla_i f(w_t)]_j) \neq \text{sign}(u_t^{(i,j)}))$$

This follows from the fact that $\|u_t^{(i)}\|_m \leq \sqrt{\frac{2\log(d_i)}{1-\beta_2}}$ and $\sqrt{v_t} \leq G$. If $\beta_2 = 0$, then $T_1$ can be bounded as follows:

$$T_1 \leq -\eta_t \sum_{i=1}^{h} \sum_{j=1}^{d_i} \sqrt{\frac{1}{2 \log(d_i)}} \left( \|w_t^{(i)}\|_m \times [\nabla_i f(w_t)]_j \right)$$

$$-\eta_t \sum_{i=1}^{h} \sum_{j=1}^{d_i} \left( \|w_t^{(i)}\|_m \times [\nabla_i f(w_t)]_j \times \frac{u_{t,j}^{(i)}}{\|u_t^{(i)}\|_m} \mathbf{1}(\text{sign}([\nabla_i f(w_t)]_j) \neq \text{sign}(u_{t,j}^{(i)})) \right)$$

The rest of the proof for $\beta_2 = 0$ is similar to argument for the case $\beta_2 > 0$, which is shown below. Taking expectation, we have the following:

$$\mathbb{E}[T_1] \leq -\eta_t \sum_{i=1}^{h} \sum_{j=1}^{d_i} \sqrt{\frac{1-\beta_2}{G^2 2 \log(d_i)}} \mathbb{E}\left[\|w_t^{(i)}\|_m \times \left([\nabla_i f(w_t)]_j \times g_{t,j}^{(i)}\right)\right]$$

$$-\eta_t \sum_{i=1}^{h} \sum_{j=1}^{d_i} \mathbb{E}\left[\|w_t^{(i)}\|_m \times \left([\nabla_i f(w_t)]_j \times \frac{u_{t,j}^{(i)}}{\|u_t^{(i)}\|}\right) \mathbf{1}(\text{sign}([\nabla_i f(w_t)]_j) \neq \text{sign}(g_{t,j}^{(i)}))\right]$$

$$\leq -\eta_t \sum_{i=1}^{h} \sum_{j=1}^{d_i} \sqrt{\frac{1-\beta_2}{G^2 2 \log(d_i)}} \mathbb{E}\left[\left(\|w_t^{(i)}\|_m \times [\nabla_i f(w_t)]_j \times g_{t,j}^{(i)}\right)\right]$$

$$+\eta_t \sum_{i=1}^{h} \sum_{j=1}^{d_i} \mathbb{E}\left[\alpha_u |[\nabla_i f(w_t)]_j| \mathbf{1}(\text{sign}([\nabla_i f(w_t)]_j) \neq \text{sign}(g_{t,j}^{(i)}))\right]$$

Using the bound on the probability that the signs differ, we get:

$$\mathbb{E}[T_1] \leq -\eta_t \alpha_l \sqrt{\frac{h(1-\beta_2)}{G^2 2 \log(d)}} \|\nabla f(w_t)\|^2 + \eta_t \alpha_u \sum_{i=1}^{h} \sum_{j=1}^{d_i} \frac{\sigma_{i,j}}{\sqrt{b}}.$$

Substituting the above bound on $T_1$ in equation 6, we have the following bound:

$$\mathbb{E}[f(w_{t+1})] \leq f(w_t) - \eta_t \alpha_l \sqrt{\frac{h(1-\beta_2)}{2G^2 \log(d)}} \|\nabla f(w_t)\|^2 + \eta_t \alpha_u \frac{\|\tilde{\sigma}\|_1}{\sqrt{b}} + \frac{\eta_t^2 \alpha_u^2 d \|L\|_1}{2}$$

Summing the above inequality for $t = 1$ to $T$ and using telescoping sum, we have the following inequality:

$$\mathbb{E}[f(w_{T+1})] \leq f(w_1) - \eta_t \alpha_l \sqrt{\frac{h(1-\beta_2)}{2G^2 \log(d)}} \sum_{t=1}^{T} \mathbb{E}[\|\nabla f(w_t)\|^2] + \eta T \alpha_u \frac{\|\tilde{\sigma}\|_1}{\sqrt{b}} + \frac{\eta^2 \alpha_u^2 dT}{2} \|L\|_1.$$

Rearranging the terms of the above inequality, and dividing by $\eta T \alpha_l$ we have:

$$\sqrt{\frac{h(1-\beta_2)}{2G^2 \log(d)}} \frac{1}{T} \sum_{t=1}^{T} \mathbb{E}[\|\nabla f(w_t)\|^2] \leq \frac{f(x_1) - \mathbb{E}[f(w_{T+1})]}{T \eta \alpha_l} + \frac{\alpha_u \|\tilde{\sigma}\|_1}{\alpha_l \sqrt{b}} + \frac{\eta d \alpha_u^2}{2\alpha_l} \|L\|_1$$

$$\leq \frac{f(w_1) - f(w^*)}{T \eta \alpha_l} + \frac{\alpha_u \|\tilde{\sigma}\|_1}{\alpha_l \sqrt{b}} + \frac{\eta d \alpha_u^2}{2\alpha_l} \|L\|_1$$

