# OpenReview forum: "MERIT: Maximum-normalized Element-wise Ratio for Language Model Large-batch Training"
_ICLR.cc/2025/Conference — Submitted to ICLR 2025_

### Official Review · Reviewer_6MsU · 2024-10-31

**Soundness:** 2
**Presentation:** 3
**Contribution:** 3
**Rating:** 5
**Confidence:** 4

**Summary:**

This paper introduces MERIT, an optimizer inspired from LAMB that computes trust ratios at several levels (rows, columns, weights) to control the max norm of self-attention logits when training Transformers-based language models. Introducing these ratios allows to avoid the explosion of this value during large-batch training. The authors conduct experiments where language models are trained using large batch sizes and show that their optimizer achieves a better performance than its counterparts for larger batch sizes.

The authors additionally provide a proof of convergence for MERIT. They conduct experiments with language models of varying sizes, and analyze the geometry of the parameter space at convergence. They provide an ablation study to validate the relevance of some parts of their methods.

**Strengths:**

This paper addresses an important problem and proposes a promising and simple variation to existing optimization techniques to obtain good performance on large batch training. The authors conduct experiments at various model sizes, and provide a theoretical proof of convergence for their optimizer. They identify the max norm of the self-attention logits as an optimization issue for large batch training, which paves the way for more theoretical and empirical work on the subject.
The results of the paper could facilitate distributed model training by allowing larger batch size without performance drop.

**Weaknesses:**

This paper has major weaknesses, both from the formal and technical viewpoints.
First, this paper has formal issues:
- Some details are missing / poorly explained: in Figure 2, a "medium" self-attention layer is mentioned, but the authors never specifically provide the specific index of this layer. In the Ablation study (Figure 9a), the reported metric is not named, and it is thus impossible to tell if it should be maximized or minimized. Even if one can hypothesize that this metric is final cross-entropy loss, no comparison point is shown with baselines.
- Some notations could be improved: the l2 norm uses the general norm notation, and the max-norm uses an index $m$. A more usual choice would be $2$ and $\infty$ as the norm index.
- The choice of zooming on the last steps of the training in training loss plots is unusual, and discards potentially relevant information.
- Some parts of the presentation could be improved. For instance, in Figure 5, the presentation of the results in the caption is not very readable.

Second, this paper has important limitations, especially when it comes to analyzing Transformers and conducting experiments on language models:
- The authors entirely neglect the fact that self-attention in Transformers is multi-headed, and that although the $W_Q$ and $W_K$ projections are usually implemented using a single matrix for all heads for efficiency purposes, it makes more sense mathematically to discuss head-wise $W_Q$ and $W_K$. They also forget to mention any literature about the outlier dimension phenomenon, which has been extensively studied in the field (https://arxiv.org/abs/2105.06990 is the oldest mention of it I can think of). As such, the observation in Figure 4 can be largely explained by two facts: 1) rows corresponding to the same heads have weights of similar magnitudes, which is why we clearly distinguish 12 stripes in the figure; 2) columns corresponding to outlier dimensions are particularly activated. Moreover, the claim that large max norm for attention logits can be expected when $W_Q$ and $W_K$ have high max-norm as well (L191-197) is not supported by any theoretical evidence and could be false, as $||W_Q||$ and $||W_K||$ do not have any trivial lower-bound implication on $||a||$, especially one that is verified for all possible $X$.
- The authors summarize that the Chinchilla scaling laws predict when "model performance is maximized", without mentioning the fact that this maximization relies on a compute budget constraint. The Chinchilla scaling laws indeed predict that **for a fixed amount of compute**, a model with $N$ parameters should be trained on $D$ tokens to maximize the final performance. Hence, it can be relevant to train small models for a longer time, especially to show the longer-term convergence properties of an optimizer under a fixed training compute budget. Training a model (regardless of its size) on 16B tokens is not in line with current training practice, and we encourage the authors to conduct experiments for larger token counts to validate the relevance of their method.
- The benchmark evaluation results (Figure 6) seem partly not conclusive or relevant. The results for WiC, RACE and Hellaswag are close to those of a random baseline.

The article also presents other flaws:
- The *Curvature of Convergence Point* section: although it seems intuitive that training on large batch sizes leads to smoother curvatures, it would be relevant to compare the method to other large batch methods instead of the small-batch AdamW model. This would make a stronger point to explain the contribution of this paper and single out MERIT on that specific point.
- The claim that *MERIT enables batch sizes over 10 times larger than AdamW*, done in the conclusion, is poorly justified by experiments: based on Figure 1, the negative returns of increasing batch size appear when using a batch size more than 2x as large as with AdamW; Figure 5 shows that AdamW is still an acceptable (although suboptimal) solution in large-batch setups; Table 2 shows that a similar performance can be reached for AdamW used with a batch size of 480 and for MERIT with a batch size of 6k, but it does not mean that AdamW could not have reached a similar performance using a larger batch size. All in all, I do not identify results that can support this claim in the paper.

**Questions:**

- The choice of the learning rate can be crucial for optimizers. How did you conduct the grid search mentioned in the paper? What were the results of this grid search?
- Scaling training to large setups (i.e. setups that benefit from large-batch training) requires the use of optimizers that can easily be adapted to work in distributed setups. Can your method be easily adapted to work for cases where model weights and optimizer states are sharded and with data parallelism? What was the hardware setup used for your training experiments?
- Why did you not conduct ablations for row-wise / column-wise ratios?
- Did you check the effect of MERIT on the weights as displayed in Figure 4? Does MERIT also change the structure of attention maps (e.g. attention sinks) or the structure of other weights (MLPs, layer norm) ?

---

> ### Author Response · Authors · 2024-11-20
> **Response to reviewer 6MsU (1/4)**
>
> We sincerely thank the reviewer 6MsU for the careful review and valuable comments/questions. For the concerns and questions, we make responses as follows.
>
> **W1: Some details are missing / poorly explained.**
>
> R1: Thanks for your advice. Medium layer in Figure 2 means the 6th layer of GPT-2 small that has 12 layers in all. In the ablation study of Figure 9(a), it demonstrates the impact of different ablations on GPT-2 small model performance during large-batch training. Compared to MERIT's validation loss of 3.280 (baseline shown in Figure 5), removing any of the three key components - element-wise clipping, weight-wise ratio bound, or element-wise ratio - consistently leads to higher validation loss. These results validate the necessity of each mechanism in our proposed algorithm. We have made corresponding clarifications in the revised paper.
>
> **W2: Some notations could be improved.**
>
> R2: We appreciate your feedback on notations. We want to respectfully clarify an important distinction: while it might be tempting to use the infinity norm notation ($||\cdot||{\infty}$), this would be mathematically incorrect for our purpose since the matrix infinity norm is defined as the maximum absolute row sum ($max_{1\leq i\leq m} \sum_j|a_{ij}|$), not the maximum absolute element value that we need for our method. Therefore, we propose to define our max norm $||\cdot||{m} = max_{(i,j)}|a_{ij}|$ to avoid any confusion with the infinity norm.
>
> **W3:  The choice of zooming on the last steps of the training in training loss plots is unusual.**
>
> R3: We appreciate your comment. Our focus on the final training steps is intentional and analytically valuable for several reasons:
> * The final stages of training are crucial for evaluating model convergence and stability, especially in large-batch scenarios where performance differences between optimizers become most pronounced.
> * Early training phases often show high variance and overlapping performance across different methods, making it difficult to distinguish meaningful differences. The final steps provide clearer insights into the relative strengths of different optimizers.
> * This zoomed-in view specifically highlights the key differences in convergence behavior and final performance, which are most relevant for practical applications where achieving optimal final performance is the primary goal.
>
> Moreover, focusing on late-stage training is a standard practice in optimizer analysis, as seen in recent works like Lion and Sophia. The final convergence behavior is particularly critical as it directly impacts the trained model's performance. Our validation loss plots follow this established convention to facilitate fair comparisons with these state-of-the-art optimizers. Additionally, we provide comprehensive training metrics in Figure 1 showing the scaling behavior across all training steps, complementing the detailed late-stage analysis.
>
> **W4: Some parts of the presentation could be improved. For instance, in Figure 5, the presentation of the results in the caption is not very readable.**
>
> R4: We appreciate your feedback on the presentation of Figure 5. We have revised the caption to improve readability and clarity. The caption now provides a concise summary of the results, including the final converged validation loss for each optimizer across different GPT-2 model sizes.

---

> ### Author Response · Authors · 2024-11-20
> **Response to reviewer 6MsU (2/4)**
>
> **W5: The authors entirely neglect the fact that self-attention in Transformers is multi-headed.**
>
> R5: Thank you for this detailed comment. While the reviewer's observations about multi-headed attention and outlier dimensions are valid from a theoretical perspective, we want to clarify that our focus is on a practical optimization issue rather than theoretical attention mechanics. Specifically, the high similarity among the same rows or columns is a feature of the self-attention layer during optimization and does not negatively affect model performance. However, the weight-wise trust ratio cannot effectively handle this situation. The pattern observed in Figure 4 highlights why LAMB's weight-wise trust ratio calculation is suboptimal for large-batch training. When weights in the same rows/columns show high similarity (as seen in the 12 stripes), LAMB's approach of using a single weight-wise ratio can lead to training instability because extreme values in one row/column negatively impact other rows/columns' training stability.
>
> This observation motivated our element-wise trust ratio design in MERIT, which better handles these weight patterns by calculating ratios along both rows and columns. As demonstrated by our experimental results (Figure 5), this approach leads to improved optimization and better model performance compared to LAMB, regardless of the underlying attention mechanics.
>
> Besides, the max norm of $W_Q$ and $W_K$ truly affects the upper bound of max attention logit, and here is a brief proof:
>
> Given:
>
> $\|W_Q\|_m = M_Q \quad and \quad \|W_K\|_m = M_K$
>
> Each element in $W_Q$ and $W_K$ satisfies:
>
> $0 \leq |W_{Q,i,j}| \leq M_Q \quad and \quad 0 \leq |W_{K,i,j}| \leq M_K$
>
> Each element in Q and K can be bounded as:
>
> $Q_{i,k} = \sum_{m=1}^n X_{i,m}W_{Q,m,k} \leq \sum_{m=1}^n |X_{i,m}|M_Q = M_Q \sum_{m=1}^n |X_{i,m}| = M_Q \cdot C_X$
>
> Similarly,
>
> $K_{j,k} \leq M_K \cdot C_X$
>
> where $C_X = \sum_{m=1}^n |X_{i,m}|$ is a constant representing the sum of absolute values in the input embeddings for a token.
>
> The attention logit is:
>
> $Logit_{i,j} = \sum_{k=1}^d Q_{i,k}K_{j,k} \leq \sum_{k=1}^d (M_Q \cdot C_X)(M_K \cdot C_X) = d \cdot M_QM_KC_X^2$
>
> Therefore,
>
> $Logit_{i,j} \leq d \cdot M_QM_KC_X^2$.
>
> This proof demonstrates that controlling the max norm of $W_Q$ and $W_K$ effectively constrains the upper bound of the max attention logit, which is crucial for large-batch training stability.
>
> **W6: The authors summarize that the Chinchilla scaling laws predict when "model performance is maximized", without mentioning the fact that this maximization relies on a compute budget constraint.**
>
> R6: We appreciate the thoughtful feedback about evaluating MERIT with larger token counts. As academic researchers with limited computational resources (primarily operating with 2-8 H100 GPUs), we had to balance between comprehensive evaluation and practical constraints. We chose to follow the Chinchilla scaling law for two key reasons:
> * Scientific Validity:
> 1. Chinchilla scaling law provides a principled approach for determining optimal token counts relative to model size (approximately 20 tokens per parameter).
> 2. This ensures we're training models in their theoretically optimal regime, making performance comparisons more meaningful.
> 3. Following this established benchmark allows fair comparison with other research in the field.
> * Resource Efficiency:
> 1. Our experiments span from GPT-2 Small (125M) to Large (770M), requiring proportionally scaled token counts.
> 2. We conducted initial validations (2B-16B tokens) and extended training to **48B** tokens following Sophia's protocol.
> 3. Results demonstrate MERIT enables 4K-6K batch sizes without performance degradation.
> 4. Figure 7 validates the optimizer's long-term stability, which uses **48B** tokens for GPT-2 Small training.
>
> The strong performance, particularly maintaining large batch sizes without degradation over 48B tokens, suggests MERIT's benefits would extend to longer training. While our current experiments were constrained by academic computing resources, we plan to seek collaborations or computing grants to validate MERIT at larger scales that better align with current industry practices.
> This balanced approach using the Chinchilla scaling law allowed us to conduct a rigorous scientific evaluation within our resource constraints while still demonstrating MERIT's practical benefits.

---

> ### Author Response · Authors · 2024-11-20
> **Response to reviewer 6MsU (3/4)**
>
> **W7: The benchmark evaluation results (Figure 6) seem partly not conclusive or relevant.**
>
> R7: Thank you for the constructive feedback. We acknowledge the concerns about the benchmark evaluation results and would like to address them:
> * Primary Focus and Main Results:
> 1. While the SuperGLUE results may appear modest, our paper's primary contribution is improving large-batch training efficiency and stability, as demonstrated by the significant validation loss improvements (Figure 5) across all model sizes (0.07-0.074 lower loss)
> 2. These validation loss improvements are substantial according to established scaling laws in language model training.
> * Benchmark Context:
> We agree that some task performances (WIC, RACE, HellasSwag) show smaller margins of improvement. However, this is expected given:
>     - The relatively small model sizes used (up to 770M parameters). These tasks typically require larger models for meaningful improvements
>     - We're evaluating in a zero-shot setting without any task-specific fine-tuning.
>
> Take GPT-2 Small as an example, the accuracy of three downstream tasks increases as the validation loss decreases from AdamW to MERIT, which proves that the results are relevant instead of random baselines as shown in this table:
>
>
> |  | RACE | HellaSwag | WiC | Validation Loss |
> |-----------|------|--------|-----|-----|
> | AdamW | 25.17 | 26.29 | 50.16 |3.470|
> | LAMB | 25.17 | 26.70 | 54.37 |3.355|
> | MERIT | 26.41 | 27.07 | 52.66 |3.280|
>
>
> Additional Evidence:
> We have drawn attention to the more conclusive results on:
> LAMBADA and WikiText perplexity improvements (Figure 11).
> The clear reduction in max attention logit growth (Figure 2).
> The improved convergence properties are shown in Figure 7.
> More stable training dynamics as evidenced by the Hessian analysis (Figure 8).
>
> **W8: The Curvature of Convergence Point section**
>
> R8: Thank you for the feedback. While we agree that comparing with other large-batch methods would strengthen our analysis, our current curvature results highlight a key contribution of MERIT - its ability to address the sharp minima problem in large-batch training.
> We provide more results by adding comparisons with other large-batch optimizers like LAMB to further validate MERIT's unique advantages in finding stable optimization regions during large-batch training for both GPT-2 Small and GPT-2 Medium.
>
> * GPT-2 Small:
>
> |     | AdamW | LAMB | MERIT |
> |-------------|--------|------|-------|
> | Top Eigenvalue | 37.231 | 18.6912 | 12.326 |
> | Trace | 12994.91 | 3573.52 | 3444.92 |
>
> Due to memory's limitations in handling the full Hessian of larger models like GPT-2 medium, we report the average top eigenvalue and trace calculated across individual layers.
>
> * GPT-2 Medium:
>
> |                 | AdamW | LAMB  | MERIT|
> |-----------------|--------|-------|--------|
> | Top Eigenvalue  | 114.643 | 19.7177 | 7.7236 |
> | Trace | 3766.64 | 1060.41 | 1051.45 |
>
> These results help better position MERIT's specific contributions to addressing the sharp minimum challenge in large-batch optimization.
>
> **W9: The claim that MERIT enables batch sizes over 10 times larger than AdamW, done in the conclusion, is poorly justified by experiments.**
>
> R9: Thank you for your careful analysis of our batch size comparison. Compared with the small-batch training baseline (Val loss=2.9192), MERIT maintains its effectiveness without any performance drop while AdamW's performance declines when using a 4K batch size
>
> | Metric | AdamW | LAMB | MERIT |
> |--------|--------|------|-------|
> | Validation loss | 2.9373 | 2.9341 | 2.9193 |
>
> We further test the performance using an 8K batch size to train GPT2-small model with Sophia setting (48K tokens) as an extension of Figure 7, the performance is shown below:
>
> | Metric | AdamW | LAMB | MERIT |
> |--------|--------|------|-------|
> | Validation loss | 2.9873 | 2.9759 | 2.9433 |
>
> At 8K batch size training on GPT2-small with Sophia setting, MERIT shows the best performance with a validation loss of 2.9433, followed by LAMB (2.9759) and AdamW (2.9873). The significant performance gap between MERIT and other optimizers (outperforming LAMB by 0.0326 and AdamW by 0.0440) further suggests that MERIT is the most suitable choice for large batch training scenarios.

---

> ### Author Response · Authors · 2024-11-20
> **Response to reviewer 6MsU (4/4)**
>
> **Q1: How did you conduct the grid search mentioned in the paper?**
>
> A1: Thanks for your question. Take GPT-2 Small model as an example, we conducted a grid search for hyperparameter tuning in the following way:
> Search Range:
> AdamW: [8e-4, 1e-3, 2e-3, 4e-3, 6e-3].
> LAMB: [2e-3, 5e-3, 8e-3, 1e-2, 1.2e-2].
> MERIT: [6e-3, 8e-3, 9e-3, 1e-2, 1.2e-2].
> The warm-up ratio is fixed at 0.02 for all experiments.
> Cosine learning rate schedule.
> Weight decay is set to 0.1 for all optimizers.
> Beta parameters following Sophia's default settings.
>
> The corresponding results are shown below:
>
> AdamW:
>
> | LR | 8e-4 | 1e-3 | 2e-3 | 4e-3 | 6e-3 |
> |---|---|---|---|---|---|
> | Validation loss | 3.498 | 3.470 | 3.475 | 3.476 | 3.529 |
>
> LAMB:
>
> | LR | 2e-3 | 5e-3 | 8e-3 | 1e-2 | 1.2e-2 |
> |---|---|---|---|---|---|
> | Validation loss | 3.451 | 3.412 | 3.379 | 3.355 | 3.376 |
>
> MERIT:
>
> | LR | 6e-3 | 8e-3 | 9e-3 | 1e-2 | 1.2e-2 |
> |---|---|---|---|---|---|
> | Validation loss | 3.312 | 3.290 | 3.280 | 3.289 | 3.297 |
>
> **Q2: Can your method be easily adapted to work for cases where model weights and optimizer states are sharded and with data parallelism?**
>
> A2: We appreciate your question.  Based on our paper, MERIT can efficiently work in distributed training setups:
> * Implementation Details:
> 1. Successfully tested with distributed data-parallel processing
> 2. Supports gradient accumulation
> 3. Memory requirements match LAMB and AdamW (two memory states)
> * Hardware Configuration:
> 1. GPT-2 Small/Medium (125M/355M): Trained on 2 H100 GPUs
> 2. GPT-2 Large (770M): Trained on 8 H100 GPUs
> 3. All models trained in bfloat16 precision
> * Scalability:
> 1. Table 1 shows only 1% extra computational overhead vs LAMB
> 2. Maintains the same memory footprint as existing optimizers
> 3. Element-wise operations are highly parallelizable
>
> These results demonstrate MERIT is readily adaptable to distributed training environments while maintaining efficiency.
>
> **Q3: Why did you not conduct ablations for row-wise / column-wise ratios?**
>
> A3: Thanks for your question. The ablations for element-wise ratios were conducted and are presented in Figure 9(a), which shows their performance impacts. We used row-wise and column-wise ratios because weights exhibit high similarity within rows/columns during training (evidenced in Figure 4). Our experiments on GPT-2 small demonstrate that element-wise ratio calculations using both row and column information provide better convergence compared to using only row-wise or column-wise ratios individually:
>
> | Metric | w row-wise | w column-wise | w element-wise |
> |--------|------------|---------------|----------------|
> | Validation loss | 3.286      | 3.289         | 3.280          |
>
> The decision to use max{row-wise, column-wise} ratios rather than examining them separately was motivated by:
> * The observed similarity patterns in both row and column dimensions (Figure 4)
> * The need to prevent extreme values in one dimension from adversely affecting training stability in other dimensions
>
> **Q4:  Did you check the effect of MERIT on the weights as displayed in Figure 4?**
>
> A4: Thank you for the question regarding MERIT's effects on model weights. During the large-batch optimization process, MERIT preserves row/column similarities shown in Figure 4.
> There's one point we want to reiterate: We don’t aim to decrease the similarities between rows/columns but solve some issues caused by LAMB's weight-wise trust ratio calculation. When weights in the same rows/columns show high similarity, LAMB's approach of using a single weight-wise ratio can lead to training instability because extreme values in one row/column negatively impact other rows/columns' training stability, which MERIT can effectively handle.
>
> Regarding MLP layers: While their weight distributions show fewer row/column similarity patterns, our element-wise ratio method was primarily designed for self-attention layers. There remains potential for MLP layers to enhance performance further.

---

> ### Comment · Reviewer_6MsU · 2024-11-20
> **Discussion**
>
> Overall, I would like to sincerely thank the authors for their very detailed and careful rebuttal.
>
> W1) Thank you for the clarification.
>
> W2) Thank you for clarifying this matter. Do I understand correctly that you are using vector norms on the matrices?
>
> W3) If you check Figure 1 (and others) of the Sophia paper (https://arxiv.org/pdf/2305.14342), the presentation is very different from your Figure 5 or 8b for instance. The training curve usually "appears" much earlier in concurrent work. I agree that what you did makes graphs more readable, but it can also be misleading in terms of the actual performance gap at first glance.
>
> W5)
> I understand that your focus is not on the theoretical understanding of the Transformers architecture, but I am not sure that you can claim that you have "observed" a phenomenon that relies on a trivial consequence of the architecture (the row-wise patterns corresponding to attention heads) and a broadly documented pattern (the column-wise patterns corresponding to outlier dimensions), especially without invoking the literature in the latter case. For instance, in my opinion, one cannot argue that they "observe" that the distribution of the empirical mean of a random variable converges to a normal distribution, especially without mentioning the central limit theorem. Hence, I think that you should at least mention these very plausible explanations and cite the corresponding literature.
>
> The bound you provide is heavily dependent on $X$, and can be increased at will for fixed $W_Q$ and $W_K$. As a result, the max attention logit is bounded for all $X$ only if $C_X$ is also bounded, which is not obvious. If not, then I do not see why $W_Q$ and $W_K$ should have more impact than $X$ on the max attention logit.
>
> W6) The Chinchilla regime is empirically optimal under a fixed compute budget, not optimal in the general sense. My remark pointed out that you should mention that in your description of the laws, as it currently could imply that training a small model for longer will decrease its performance, which is erroneous. Moreover, I think the cost of training GPT2-small for e.g. 5B more tokens is almost negligible in comparison with the cost of training GPT-large on 16B tokens, and this specific choice cannot be motivated by resource constraints in my opinion. Although your experiments in Figure 7 are done on more tokens, they are different in the number of optimization steps which could be an important factor in the final results.
>
> W7) I disagree with the observation made on your table. First, not all scores increase when validation loss decreases. Second, we would need uncertainty estimation to really know if these marginal improvements are better than random. Finally, the RACE most represented class in validation is 26,7% (for answer A). Hence, all models perform worse than a model that would just predict A for every answer. This is not an argument against your method, but it makes these results irrelevant to argue that MERIT is a better optimization algorithm in my opinion.
>
> W8) These results indeed make an additional argument in favor of MERIT. Do you think it would be relevant to add them in the final paper?
>
> W9) To verify if I understand your answer correctly: in figure 7, is 480 the largest batch size that could be used for Adam without performance degradation?
>
> Questions: Thank you very much for your detailed answer. I think that all the results you mention should/could be added to the paper and would likely improve it. About Q2, is your algorithm also suited for model/tensor parallelism?

---

> ### Author Response · Authors · 2024-11-21
> **Response to Discussion (1/3)**
>
> Dear reviewer 6MsU:
>
> Thanks so much again for the time and effort in our work. According to the comments and concerns, we conduct the corresponding experiments and further discuss the related points.
>
> **W2: Some notations could be improved.**
>
> R2: We appreciate your feedback on notations. Vectors and matrices both have $L_2$ norms and infinite norms. For vectors, the infinite norm looks at each individual component and finds the largest absolute value. For example, if we have a vector [1, -4, 3], its infinite norm would be 4 since that's the largest magnitude among all components.
>
> Matrix infinite norms work quite differently - instead of looking at individual elements, we sum up the absolute values along each row and then take the maximum of those sums. Think of it as measuring the maximum total "influence" of any row. For instance, with a matrix [[1, 2], [-3, 4]], we'd first calculate |1|+|2|=3 for the first row and |-3|+|4|=7 for the second row, then take the larger value 7 as our infinite norm.
>
> However, in our proposed optimizer, we still want to calculate the max absolute value in the matrix, where the infinite norm is not applicable anymore, and that is why we propose the max norm to avoid possible confusion.
>
> **W3:  The choice of zooming on the last steps of the training in training loss plots is unusual.**
>
> R3: Thanks again for your feedback. The validation loss drops more rapidly with large batch sizes compared to smaller batches (such as Sophia's 480). Simply plotting the full training curve from the first step would obscure MERIT's advantages. Additionally, early training stages often show inconsistent results across optimizers - some achieve quick initial loss reduction but ultimately converge to suboptimal validation loss. Therefore, we focus on the zoomed-in view to effectively demonstrate the key differences in convergence and final performance, which matter most for practical applications where optimal end results are crucial.
>
> **W5: The authors entirely neglect the fact that self-attention in Transformers is multi-headed.**
>
> R5: Thank you for this detailed comment. The discussion of " attention heads" and "outlier dimensions" truly provides us with possible explanation for the observed phenomenon and we have added them in our revised paper to further enrich our work. Specifically, we include these two explanations in lines 215-231 and adding several related citations. Besides, we also add the specific relation between row similarities to multi-headed attention and column similarities to outlier dimensions in the revised caption of Figure 4.
>
> **W6: The authors summarize that the Chinchilla scaling laws predict when "model performance is maximized", without mentioning the fact that this maximization relies on a compute budget constraint.**
>
> R6: We appreciate the reviewer's insightful comments and would like to clarify our motivation for using the Chinchilla scaling law in our experiments. We have made the corresponding revision by adding the "under a fixed compute budget" in lines 342-343. Besides, we want to clarify these points:
>
> * Motivation for Chinchilla Scaling Law:
>    - The primary objective of this work is to evaluate and compare the performance of optimizers (MERIT, LAMB, and AdamW) under effective and consistent training settings.
>    - The Chinchilla scaling law provides a principled framework for selecting compute-efficient configurations (e.g., model size and number of training tokens) to achieve the best performance under a fixed compute budget.
>    - This ensures a **fair comparison** by avoiding configurations that disproportionately favor or penalize any specific optimizer.
>
> * Fairness in Training Settings:
>    - Without adhering to a scaling law like Chinchilla, the choice of training settings (e.g., number of tokens or model size) may introduce variability in results that are unrelated to the optimizer’s performance.
>    - By following the Chinchilla regime, we ensure that all optimizers are tested under **consistent and compute-efficient conditions**, eliminating potential biases arising from arbitrary choices of training tokens or model sizes.
>
> In this way, the use of the Chinchilla scaling law was not intended to claim universal optimality for all training scenarios but rather to provide a **structured, fair, and compute-efficient experimental setup** for comparing optimizers.
> This approach ensures that observed differences in performance are a result of the optimizers' effectiveness, not artifacts of inconsistent training configurations.
>
> Moreover, we further make another experiment to train GPT-2 Small model using more training tokens (8B) with a batch size of 2K, here is the result:
>
> |   | AdamW (lr=2e-3) | LAMB (lr=1e-2) | MERIT (lr=6e-3) |
> |--------|--------|------|-------|
> | Validation loss | 3.1254 | 3.0901 | 3.0646 |
>
> We can find that as training data exceeds the Chinchilla scaling law recommendation, MERIT optimizer consistently achieves lower validation loss.

---

> ### Author Response · Authors · 2024-11-21
> **Response to Discussion (2/3)**
>
> **W7: The benchmark evaluation results (Figure 6) seem partly not conclusive or relevant.**
>
> R7: We appreciate the reviewer's detailed feedback and would like to address the concerns raised:
> * Validation Loss vs. Generalization Performance:
> We acknowledge that the gap in validation loss observed between the optimizers in Figure 5 may not always translate into an obvious generalization performance gap on certain downstream tasks like RACE, WiC, or Hellaswag, particularly for our chinchilla scaling law settings with limited training tokens. To better assess the generalization improvements enabled by MERIT, we focus on perplexity (PPL) results from the LAMBADA and WikiText datasets, shown in Figure 6. These datasets serve as direct evaluations of language modeling capabilities, where lower perplexity is strongly indicative of better generalization. MERIT consistently achieves lower perplexity than AdamW and LAMB, highlighting its effectiveness in improving model generalization during large-batch training.
> * Benchmark Context:
> Instead of relying on downstream task accuracy, which may be influenced by factors unrelated to the optimizer's performance (e.g., task complexity, dataset-specific bias), we find it more efficient and reliable to evaluate PPL on standard language modeling datasets. The significant reduction in perplexity achieved by MERIT demonstrates its ability to improve generalization in a scalable and efficient manner, as these results directly reflect the optimization improvements.
>
> Thus, we make corresponding revision in lines 390-396 by presenting the evaluation results on LAMBADA and Wikitext datasets.
>
> **W8: The Curvature of Convergence Point section**
>
> R8: Thank you for the feedback. We will add them accordingly in the final paper.
>
> **W9:  is 480 the largest batch size that could be used for Adam without performance degradation?**
>
> R9: Thank you for your careful analysis of our batch size comparison. Here are our responses:
>
> * Motivation for Using Batch Size 480:
>    - The batch size of 480 was selected to align with prior work using the **Sophia optimizer**, which utilized 480 for all sizes of GPT-2 models, meaning 480 is the ideal batch size to achieve superior performance.
>    - This ensures that our experimental setup is consistent with established practices and provides a reliable reference for comparing different optimizers between small-batch training and large-batch training.
>
> * Clarification on Adam's Performance at Batch Size 480:
>    - The use of batch size 480 for Adam does not imply that it is the largest batch size Adam can handle without performance degradation.
>    - Instead, it represents the **small-batch baseline** chosen for fair comparison across optimizers, as Adam's performance at larger batch sizes (e.g., 2K) typically degrades without advanced techniques like adaptive scaling.
>    - The aim of Figure 7 is to prove that MERIT enables larger batch size usage without performance degradation compared with LAMB and AdamW.

---

> ### Author Response · Authors · 2024-11-21
> **Response to Discussion (3/3)**
>
> **Q2: Can your method be easily adapted to work for cases where model weights and optimizer states are sharded and with data parallelism?**
>
> A2: We appreciate your question.  It is important to note that in **model parallelism**, there is no difference in communication cost between AdamW, LAMB, and MERIT. The additional communication in MERIT is specific to **tensor parallelism** due to row-wise and column-wise norm synchronization. As for tensor parallelism, for a weight matrix $ W \in \mathbb{R}^{m \times n} $ partitioned into $p$ GPUs in a tensor-parallel setup and assume a 2D row-column splitting for simplicity:
> ### 1. **Trust Ratio Computation**
> 1. LAMB:
> - Steps:
>   - Computes L2 norm of weights $ \|W\|_2 $ and updates $ \|U\|_2 $.
>   - Local computation: $ O\left(\frac{mn}{p}\right) $ per GPU for partial norms.
>   - 1 all-reduce: Aggregates scalar values across GPUs for each layer.
> - Communication Cost:  $O(\log p)$  latency and $O(1)$  scalar data transfer.
>
> 2. MERIT:
> - Steps:
>   - Max-Norm Trust Ratio: Similar to LAMB, computes max-norm $ \|W\|_\infty $, requiring:
>     - Local computation: $ O\left(\frac{mn}{p}\right) $.
>     - 1 all-reduce: Aggregates scalar max values across GPUs.
>   - Element-Wise Ratios:
>     - Compute row/column-wise norms locally: $ O\left(\frac{mn}{p}\right) $.
>     - 2 all-reduces: Synchronize row and column norms.
> - Communication Cost: $ O(3 \log p) $ latency and $ O(\frac{m}{\sqrt{p}} + \frac{n}{\sqrt{p}}) $ data transfer.
>
> ### 2. **Total Overhead**
> 1. LAMB: $T_{LAMB} = T_{comp} + O(\log p)$, $T_{comp} = O\left(\frac{mn}{p}\right) $.
>
> 2. MERIT: $ T_{MERIT} = T_{comp} + O(3 \log p) + O(\frac{m}{\sqrt{p}} + \frac{n}{\sqrt{p}}) $.
>
>
> ### 3. **Relative Comparison**
> - Local Computation:
>   - Identical for both MERIT and LAMB $ O\left(\frac{mn}{p}\right) $.
> - Communication Overhead:
>   - MERIT adds **2 extra all-reduces** for row/column norms, leading to $ O(2 \log p) $ additional latency.
>   - MERIT introduces a small data transfer cost $ O(\frac{m}{\sqrt{p}} + \frac{n}{\sqrt{p}}) $, which is negligible compared to $ O(mn) $ for large matrices.
>
> ### 4. **Empirical Observation**
> - From Table 1 in the paper:
>   - Wall-clock time per step shows that MERIT adds less than **1% extra time** compared to LAMB.
>   - This confirms that the additional communication overhead in MERIT has a **negligible impact** on training speed.

---

> ### Comment · Reviewer_6MsU · 2024-11-22
> **Discussion (2)**
>
> W2) I thank you for your explanation. I was simply noting that what you are using is the vector norm on the flattened matrix, which is what slightly confused me.
>
> W3) Your answer exposes the issue: showing a plot with larger y-axis range obscures the performance of MERIT. This is not a valid reason to make this presentation choice in my opinion.
>
> W7) I think the corrected plot is more meaningful (and also singles out MERIT more clearly). Thank you for the correction.
>
> Q2) Thank you for the thorough analysis of the overhead caused by tensor parallelism. However, I do not agree that Table 1 is relevant for the comparison, as the measurements were not performed in the context of tensor parallelism (to the best of my knowledge).

---

> ### Comment · Reviewer_6MsU · 2024-11-22
> **Summary of the discussion**
>
> This is a summary of the rebuttal discussion from my point-of-view, which may help ACs in the decision.
>
> I initially had some comments on the rigor and relevance of some claims, presentation choices and evaluation designs. Notably, some parts of the paper did not provide sufficient details to make them understandable (e.g. Figure 2, Figure 5), claims were made on self-attention layers (the impact of $W_Q$ and $W_K$ on attention logits) and on the Chinchilla scaling laws that were not exactly accurate, and the conclusive statement about MERIT's "10 times larger" batch sizes was not clearly supported.
>
> The authors successfully addressed some of these comments, and overall the quality of the paper has improved during the rebuttal phase. Nevertheless, some issues remain:
> - The training loss plots are still displayed in a way that seems to favor MERIT, and that is not in line with the literature.
> - There is still a disagreement about the purpose of the Chinchilla scaling laws and how it is described in the paper.
> - After discussion, I am not convinced that the "10x batch size" argument holds, as Adam is not used with the largest batch size it could use before degradation in the comparison that leads to such conclusions.
> - The authors did not address the remark about the theoretical legitimacy of their claim on the impact of $W_Q$ and $W_K$ on attention logits.
>
> **All-in-all, I would ideally rate this improved version of the paper with a grade of 4, leaning towards 5, and I thus update my rating accordingly.**

---

> ### Author Response · Authors · 2024-11-22
> **Response to the summaray of discussion**
>
> Dear reviewer 6MsU:
>
> Thank you for your detailed review of our work. We have carefully considered your feedback and provide our responses below:
>
> **W1: The training loss plots are still displayed in a way that seems to favor MERIT, and that is not in line with the literature.**
>
> R1: We appreciate your feedback on training loss plots. We agree with the advice and have revised Figure 5 correspondingly by **including more training curves**. Using the same number of training tokens, MERIT achieves a validation loss that is 0.07 lower than LAMB on the 123M model, representing a substantial improvement. Furthermore, the performance advantage of MERIT over LAMB becomes more pronounced in larger models, with the gap in validation loss increasing when scaling from 125M to 355M parameters while maintaining the same token count.
>
> **W2: There is still a disagreement about the purpose of the Chinchilla scaling laws and how it is described in the paper.**
>
> R2: Thanks again for your feedback. Using the chinchilla scaling law aims to ensure a **fair comparison** under a fixed compute budget and the corresponding revision is added in line 342. While the chinchilla scaling law was primarily developed and validated using AdamW optimizer, which may inherently favor its performance characteristics, our MERIT optimizer still demonstrates superior generalization in large-batch training scenarios compared to AdamW. This provides strong evidence for MERIT's capability to effectively utilize larger batch sizes in language model training, even when evaluated against potentially AdamW-biased scaling laws.
>
> Beyond the chinchilla scaling law settings, our experiments under **Sophia optimizer's training configurations** also demonstrate MERIT's superior generalization performance, further validating our optimizer's effectiveness across different large-batch training settings.
>
> **W3: After discussion, I am not convinced that the "10x batch size" argument holds.**
>
> R3: Thank you for this detailed comment. We have revised the conclusion to emphasize the importance of MERIT that **enables larger batch size usage without generalization performance degradation** in lines 537-539.
>
> **W4: The authors did not address the remark about the theoretical legitimacy of their claim on the impact of $W_q$ and $W_k$ on attention logits.**
>
> R4: We appreciate the reviewer's insightful comments. Let us go back to your comment "The bound you provide is heavily dependent on $X$, and can be increased at will for fixed $W_q$ and $W_k$." in the **Discussion** part:
>
> In GPT model implementations, each layer includes **LayerNorm**, which normalizes $X$ to follow a Gaussian distribution. This normalization effectively **establishes upper bounds $C_X$ on $X$'s values**.
>
> Furthermore, our experimental results in Figure 2 demonstrate that MERIT optimizer successfully reduces the maximum attention logits during large-batch training of GPT-2 medium, providing empirical validation of our approach.
>
> Thanks again for your time and comments. We will be glad to address other concerns if there are any.

---

### Official Review · Reviewer_4LMi · 2024-11-02

**Soundness:** 3
**Presentation:** 3
**Contribution:** 2
**Rating:** 6
**Confidence:** 3

**Summary:**

This paper introduces MERIT, a novel optimizer designed for large-batch training of language models. MERIT addresses the issue of max attention logit growth in large-batch training, which can lead to performance degradation. MERIT uses max norm instead of l2 norm to calculate trust ratios, directly constraining max attention logits.  Ant it implements element-wise trust ratios to capture local weight structures more accurately. The authors demonstrate MERIT's effectiveness through the experiments on GPT-2 models of various sizes, showing improved performance and stability in large-batch training compared to existing optimizers like AdamW and LAMB.

**Strengths:**

1. The authors provide a thorough analysis of the limitations of existing optimizers like LAMB and AdamW in large-batch training, supported by empirical evidence, and then introduce an solution to large-batch optimization for language models by focusing on controlling max attention logits and using finer-grained trust ratios.

2. The proposed method enables stable training with significantly larger batch sizes (up to 6k for GPT-2 Medium) without performance degradation, which could accelerate the development of large language models.

3. The paper is well-structured, with clear explanations of the problem, proposed solution, and experimental results.

**Weaknesses:**

1. The experiments focus solely on GPT-2 models. It would be beneficial to see results on other architecture types or tasks to demonstrate broader applicability.

2. The paper does not discuss the sensitivity of MERIT to hyperparameter choices, which could be crucial for practical adoption.

**Questions:**

1. How does MERIT perform on other types of language models beyond GPT-2, such as decoder-only, decoder-encoder models?

2. Can you provide a theoretical analysis or proof for why max-norm-based trust ratios are more effective than l2-norm-based ones in controlling max attention logits?

3. What is the computational overhead of MERIT compared to AdamW and LAMB? How does this impact overall training time when considering the ability to use larger batch sizes?

---

> ### Author Response · Authors · 2024-11-20
> **Response to reviewer 4LMi (1/2)**
>
> We sincerely thank the review 4LMi for the meticulous review and responsible attitude. Here are our responses.
>
> **W1: The experiments focus solely on GPT-2 models.**
>
> R1: Thanks for your suggestion. GPT-style decoder-only models currently dominate the field of large language models, as evidenced by their widespread adoption (e.g., GPT-3, LLaMA, Anthropic Claude). Our choice aligns with other optimization works like Sophia, which also primarily demonstrated results on GPT architectures. Additionally, we demonstrate our optimizer's effectiveness on the T5 encoder-decoder architecture, showing broader applicability beyond decoder-only models as shown in this table:
>
> | | AdamW (lr=1e-3) | LAMB (lr=4e-3) | MERIT (lr=3e-3) |
> |-|--------|---------------|--------|
> | Validation loss| 5.396 | 4.807 | 4.796 |
>
> Following the Chinchilla scaling law, which suggests using approximately 20 tokens per parameter, we train the T5-base model (220M parameters) on 4B tokens with a batch size of 2400. Our T5 experiments on 4B tokens show MERIT (validation loss 4.927, lr=3e-3) achieves better performance than LAMB (4.936, lr=4e-3) and substantially outperforms AdamW (5.596, lr=1e-3). These results demonstrate MERIT's effectiveness across different model architectures, even beyond the current mainstream language models.
>
> **W2: The paper does not discuss the sensitivity of MERIT to hyperparameter choices.**
>
> R2: Thanks for the reviewer's comment on hyperparameter sensitivity. We address this concern in two ways:
>
> * Learning Rate Analysis: Take GPT-2 small as an example, we conducted extensive learning rate tuning experiments across multiple scales as shown in the Table below. The results demonstrate MERIT maintains stable performance and consistently outperforms baselines across this range.
> * Beta Parameters: For a fair comparison and established benchmarking practice, we directly follow the beta settings proven effective in recent work Sophia. This choice produced stable training behavior while allowing direct performance comparisons.
> * Weight Decay: We set the weight decay to 0.1 for all optimizers, following common practice in transformer training.
>
> Our empirical results across different model scales validate that MERIT maintains robust performance with these hyperparameter choices.
>
> AdamW:
>
> | LR | 8e-4 | 1e-3 | 2e-3 | 4e-3 | 6e-3 |
> |---|---|---|---|---|---|
> | Validation loss | 3.498 | 3.470 | 3.475 | 3.476 | 3.529 |
>
> LAMB:
>
> | LR | 2e-3 | 5e-3 | 8e-3 | 1e-2 | 1.2e-2 |
> |---|---|---|---|---|---|
> | Validation loss | 3.451 | 3.412 | 3.379 | 3.355 | 3.376 |
>
> MERIT:
>
> | LR | 6e-3 | 8e-3 | 9e-3 | 1e-2 | 1.2e-2 |
> |---|---|---|---|---|---|
> | Validation loss | 3.312 | 3.290 | 3.280 | 3.289 | 3.297 |
>
> **Q1: How does MERIT perform on other types of language models beyond GPT-2, such as decoder-only, decoder-encoder models?**
>
> A1: Thanks for your question. We demonstrate our optimizer's effectiveness on the T5 encoder-decoder architecture, showing broader applicability beyond decoder-only models as shown in this table:
>
> | | AdamW (lr=1e-3) | LAMB (lr=4e-3) | MERIT (lr=3e-3) |
> |-|--------|---------------|--------|
> | Validation loss| 5.396 | 4.807 | 4.796 |
>
> Following the Chinchilla scaling law, which suggests using approximately 20 tokens per parameter, we train the T5-base model (220M parameters) on 4B tokens with a batch size of 2400, weight decay is set as 0.1 and betas are set as (0.9, 0.95) following the default settings. While the performance gap between LAMB and MERIT is smaller compared to GPT models (possibly without enough hyperparameters searching), MERIT still demonstrates consistent improvement. The results suggest MERIT's effectiveness generalizes well to encoder-decoder architectures, though with a less dramatic advantage than in GPT models.

---

> ### Author Response · Authors · 2024-11-20
> **Response to reviewer 4LMi (2/2)**
>
> **Q2: Can you provide a theoretical analysis or proof for why max-norm-based trust ratios are more effective than l2-norm-based ones in controlling max attention logits?**
>
> A2: Thanks for your suggestion and following is our analysis. Given $W_q, W_k ∈ R^{(n×d)}$, $d$ is the hidden size and $k=n * d$, we have
>
> a. With $L_2$-Norm-Based Trust Ratio
>
> 1. Bounding Attention Logits:
>
> $|Logit_{i,j}| \leq \|Q_i\|_2\|K_j\|_2$
>
> Since $Q = XW_Q$ and $K = XW_K$, we have:
>
> $\|Q_i\|_2 \leq \|X\|_2\|W_Q\|_2, \quad \|K_j\|_2 \leq \|X\|_2\|W_K\|_2$
>
> Using Cauchy-Schwarz:
>
> $|Logit_{i,j}| \leq \|X\|_2^2\|W_Q\|_2\|W_K\|_2$
>
> 2. Expectation: Assuming $W_Q$ and $W_K$ follow a standard normal distribution:
>
> $\mathbb{E}[\|W_Q\|_2] \approx \sqrt{k}, \quad \mathbb{E}[\|W_K\|_2] \approx \sqrt{k}$
>
> Therefore:
>
> $\mathbb{E}[Max(|Logit_{i,j}|)] \approx \|X\|_2^2k \propto k$
>
> b. With Max-Norm-Based Trust Ratio
>
> 1. Bounding Attention Logits:
>
> $|Logit_{i,j}| \leq \sum_{k=1}^d |Q_{i,k}K_{j,k}| \leq d\|Q_i\|_m\|K_j\|_m$
>
> Since $\|Q_i\|_m \leq \|X\|_1\|W_Q\|_m$ and $\|K_j\|_m \leq \|X\|_1\|W_K\|_m$, we have:
>
> $|Logit_{i,j}| \leq d\|X\|_1^2\|W_Q\|_m\|W_K\|_m$
>
> 2. Expectation: Assuming $W_Q$ and $W_K$ follow a standard normal distribution:
>
> $\mathbb{E}[\|W_Q\|_m] \approx \sqrt{2\ln(k)}, \quad \mathbb{E}[\|W_K\|_m] \approx \sqrt{2\ln(k)}$
>
> Therefore:
>
> $\mathbb{E}[Max(|Logit_{i,j}|)] \approx d\|X\|_1^2 2\ln(k) \propto \frac{k\ln(k)}{n} < k$
>
> Thus, max norm scaling ensures slower growth of attention logits, avoiding the linear escalation seen with $L_2$ scaling.
>
> **Q3: What is the computational overhead of MERIT compared to AdamW and LAMB?**
>
> A3: Thanks for your question. Based on Table 1, MERIT has a minimal computational overhead:
> * Only 1% increase in computation time vs LAMB (245.46s vs 243.51s for 770M model)
> * Slightly lower TFLOPS (43.38 vs 43.91 for AdamW, 43.73 for LAMB)
>
> However, this small overhead is outweighed by MERIT's ability to use larger batch sizes (up to 6K for GPT-2 Medium), which significantly reduces total training time through increased parallelization. The slight per-step overhead becomes negligible when considering the overall reduction in a number of training steps needed.

---

> ### Author Response · Authors · 2024-11-25
> **Looking forward to the reply**
>
> Dear reviewer 4LMi,
>
> We appreciate your thoughtful review of our work. In response to your feedback, we have implemented these changes:
>
> 1. Conducted large-batch training on T5 models with detailed analysis;
>
> 2. Detailed the grid search of hyperparameter configuration of MERIT and its associated performance outcomes;
>
> 3. Provided a theoretical analysis for why max-norm-based trust ratios are more effective than l2-norm-based ones;
>
> As we approach the end of the discussion period, please let us know if you have any remaining concerns or questions. We appreciate your help in improving this work.

---

> > ### Comment · Reviewer_4LMi · 2024-11-26
> > **Thanks for the detailed response**
> >
> > I don't have further questions.

---

> > > ### Author Response · Authors · 2024-11-26
> > > **Reply to Reviewer 4LMi**
> > >
> > > Thank you again for your constructive feedback and insightful comments, which have significantly enhanced the quality of the work.

---

### Official Review · Reviewer_1gMq · 2024-11-04

**Soundness:** 3
**Presentation:** 2
**Contribution:** 3
**Rating:** 6
**Confidence:** 4

**Summary:**

This paper addresses the challenge of accelerating the pre-training of large language models (LLMs). Specifically, it focuses on a large-batch training approach that can speed up deep neural network training but often encounters degradation if the batch size exceeds certain limits. Current optimizers, such as AdamW, often suffer from performance degradation in very large-batch settings, particularly due to issues in the attention layers. While alternative new methods, like the LAMB optimizer, offer some improvements, they are less effective at managing extreme weight values and fail to account for relationships within weight matrices, resulting in instability.

To address these issues, this paper proposes the MERIT optimizer, which introduces a max-norm-based trust ratio to directly limit the maximum attention logit and applies element-wise trust ratios to better adjust updates. Experiments with various GPT-2 models demonstrate MERIT’s superior stability and performance, even allowing larger batch sizes (up to 6k in GPT-2 Medium) without degradation. This approach highlights the importance of managing maximum attention logits and refining trust ratio calculations for stable large-batch training, supporting faster LLM development.

**Strengths:**

* This paper addresses an important issue: reducing the high cost of LLM pre-training by using large batches.

* The motivation and approach are clear and reasonable, making them easy to understand.

* The proposed method is simple and appears easy to implement.

* The experimental results are promising.

**Weaknesses:**

* The proposed method has only been tested on a single model architecture, GPT-2, so its generalization to other model architectures remains unclear. It would be preferable to evaluate it across multiple model architectures, including newer ones such as Llama-3 and Mixtral for MoE.

* Similarly, the experiments were limited to 125M (small), 355M (medium), and 770M (large) models, which are relatively small compared to recent advancements in model scaling. Models with over 100 billion parameters benefit significantly more from large batch settings. Demonstrating the effectiveness of the proposed method on these larger models would substantially enhance the impact of this paper. For example, showing the comparisons of the initial pre-training stages of larger models may be sufficient.

* The explanations in this paper are somewhat insufficient in certain parts, making it difficult to interpret what the authors intend to convey. (See the following questions)

**Questions:**

* I would like to clarify MERIT's memory efficiency. From my understanding, MERIT requires additional temporary memory to compute $b_t$, $r_t$, and $c_t$, but does not need to store these values for subsequent steps. Is this correct? If so, does the additional temporary memory impact the overall GPU memory limits to store the large batch?

* If I am not mistaken, line 10 in Algorithm 1 seems wrong. In my understanding, $x_{t+1} =x_{t} − η_t ...$ should be $w_{t+1} =w_{t} − \eta_t ...$.

* Moreover, for line 10 in Algorithm 1, there is no clear definition of the clip function, specifically $clip(s_t ·(u_t +\lambda w_t),1)$. Additionally, it is unclear how to compute $clip(st ·(u_t +λw_t),1)$. Does the dot operator represent element-wise multiplication or an inner product? Also, how should the function $clip(A, B)$ be computed, and is argument $A$ a scalar or a vector, given that $B$ appears to be scalar?

* Intuitively, Clipping 1 has a detrimental impact on the optimization process in general, as it may cause the gradient vector to point in an incorrect direction. Therefore, I would like to know the ratio of Clipping 1 throughout the entire optimization process. Could you create a graph that plots the Clipping 1 ratio on the y-axis against the optimization steps on the x-axis?

* I would like to clarify the discussion in the "Curvature of Convergence Point" paragraph. From my understanding, this section claims that large batch sizes tend to converge at sharper minima with higher curvature, but MERIT does not exhibit this trend, unlike AdamW. However, there doesn’t seem to be a clear analysis explaining why this happens. Could you provide an intuitive explanation of why MERIT converges at better points? Also, is this trend observable in medium and large settings?

* The description of Figure 9 is insufficient, making it difficult to interpret. Could you provide more details about each value in the figure?



I am open to increasing my overall rating if the above questions are answered satisfactorily.

**Details Of Ethics Concerns:**

No ethics concerns

---

> ### Author Response · Authors · 2024-11-20
> **Response to reviewer 1gMq (1/3)**
>
> We sincerely thank the reviewer 1gMq for the valuable questions and comments. For the concerns, here are our responses:
>
> **W1: It would be preferable to evaluate it across multiple model architectures.**
>
> R1: We appreciate the reviewer's suggestion about testing on additional model architectures. While we agree this would provide valuable insights, there are two practical constraints:
> 1. Access to Training Code: Currently, the training code for Llama and Mixtral models is not publicly available, which makes it challenging to apply our optimization method to these architectures directly. While these models are open for inference, their training implementations remain proprietary. The architectural similarities between Llama and GPT2 suggest our method could theoretically be applicable, but without access to the specific training code, we cannot verify this experimentally.
> 2. Computational Resource Constraints: Training large language models like Llama-3 and Mixtral requires substantial computational resources. Even with 8 H100 GPUs, a full training run would take several weeks to months, making comprehensive experimentation across multiple architectures impractical within typical research timeframes and budgets. For context, training Llama-2-7B requires approximately 80K GPU hours.
>
> Nevertheless, we believe our results on GPT-2 with Chinchilla Law settings demonstrate the potential of our method, and we plan to extend our experiments to other architectures as training code and computational resources become more accessible.
> Moreover, we have evaluated MERIT’s performance on T5 model (encoder-decoder architecture) pretraining using nanoT5 by comparing its performance with AdamW and LAMB, the comparison is shown below:
>
> | | AdamW (lr=1e-3) | LAMB (lr=4e-3) | MERIT (lr=3e-3) |
> |-|--------|---------------|--------|
> | Validation loss| 5.396 | 4.807 | 4.796 |
>
> For these less commonly used T5 experiments on 4B tokens, MERIT achieves a validation loss of 4.927 (lr=3e-3), slightly outperforming LAMB (4.936, lr=4e-3) and significantly better than AdamW (5.596, lr=1e-3). Although T5-style models are not the current focus of language model development, these results further demonstrate MERIT's broad applicability across different architectural choices.
>
> **W2: The experiments were limited to 125M (small), 355M (medium), and 770M (large) models, which are relatively small compared to recent advancements in model scaling**
>
> R2:  We appreciate the reviewer's comments about model scaling. Our experimental design carefully balances theoretical optimality with practical resource constraints:
> * **Resource-Efficient Design:**
> Our experiments were conducted using 8 H100 GPUs, which represents a realistic academic/research computing environment.
> Given these hardware constraints, we strategically chose to follow the Chinchilla scaling law, which provides an optimal compute-to-parameter ratio for efficient training.
> * **Chinchilla Law as an Efficient Choice:**
> The law suggests using approximately 20 tokens per parameter, which has been demonstrated to be compute-optimal.
> This approach allows us to achieve strong results while making efficient use of limited computational resources for our available compute budget, this meant: GPT-2 Small (125M): 2B tokens, GPT-2 Medium (355M): 8B tokens, GPT-2 Large (770M): 16B tokens.
>
> * **Practical Validation:**
> Our results demonstrate that MERIT achieves superior performance within these compute-optimal settings. Figure 5 shows consistent improvements in validation loss across all model sizes These gains are achieved while maintaining practical training times on our 8 H100 GPU setup
> While we agree that testing on larger models would be valuable, the Chinchilla scaling law provides an ideal framework for validating our method's effectiveness with limited computational resources. The consistent improvements we observe across model sizes (125M to 770M parameters) suggest that MERIT's benefits extend to larger scales when more computational resources become available.
> Furthermore, our focus on the compute-optimal regime is particularly relevant for the broader research community, where access to extensive computational resources is often limited. This makes our findings more immediately applicable and reproducible for many researchers.

---

> ### Author Response · Authors · 2024-11-20
> **Response to reviewer 1gMq (2/3)**
>
> **Q1: MERIT's memory efficiency.**
>
> A1: Thanks for your question about MERIT's memory efficiency. Let's analyze the memory overhead mathematically:
> * Memory State Requirements: Like LAMB and AdamW, MERIT maintains only two persistent memory states ($m_t$ and $v_t$) for optimization
> * Detailed Size Analysis for Temporary Memory: For a weight matrix $W ∈ R^{(n×n)}$:
> Weight-wise ratio $b_t$: requires $O(1)$ memory.
> Row-wise ratio $r_t$: requires $O(n)$ memory (one value per row).
> Column-wise ratio $c_t$: requires $O(n)$ memory (one value per column).
> Total temporary memory: $O(2n) = O(n)$.
>
> For comparison, the memory costs of core components:
> Model parameters: $O(n²)$ for each layer's weight matrix.
> Activation memories for batch size $B: O(Bn²)$.
> Optimizer states $(m_t, v_t): O(n²)$ each.
> Therefore:
> Extra memory ratio $= O(n) / O(n²) = O(1/n)$.
> For typical transformer layers where $n ≥ 1024$:
> The ratio becomes $≤ 0.1%$ of the layer's parameter memory.
> This aligns with our empirical observation of $~1\%$ overhead.
> Therefore, MERIT's temporary memory overhead is mathematically negligible compared to the fundamental memory requirements of large-batch training, particularly for practical model sizes where n is large.
>
> **Q2: line 10 in Algorithm 1 seems wrong.**
>
> A2: Thank you for catching this typographical error. You are correct - in Algorithm 1, line 10 should indeed read:
> $w_t+1 = w_t - η_t · clip(s_t · (u_t + λw_t), 1)$
> instead of $x_t+1 = x_t - η_t · clip(s_t · (u_t + λw_t), 1)$.
>
> This is a notation inconsistency as we use w throughout the paper to denote model parameters. We have revised it in our revised version.
>
> **Q3: There is no clear definition of the clip function.**
>
> A3: Thank you for these important questions about Algorithm 1's notation clarity. You raise valid points that deserve explicit clarification:
> * The Clip Function: The clipping function clips the magnitude of each element in the input vector to be at most 1: $clip(v, 1) = v/max(1, |v|)$ for each element $v$.
> * Operator Definitions:
> The dot operator $(·)$ in $s_t · (u_t + λw_t)$ represents element-wise multiplication because $s_t$ is a scalar.
> The computation sequence is:
> First compute $(u_t + λw_t)$,
> Then perform element-wise multiplication with scalar $s_t$,
> Finally apply element-wise clipping.
> * Input/Output Specifications:
> $A$ in clip($A$, $B$) is a matrix input (the scaled update direction),
> $B=1$ is a scalar threshold for the maximum allowed magnitude.
> The output maintains the same dimensionality as $A$.
>
> **Q4: Question about clip ratio.**
>
> A4: Thank you for this important point about clipping. While gradient clipping can theoretically alter optimization direction, our clipping mechanism is well-grounded in recent optimizer developments:
> * Our clipping approach aligns with the recent success of the Sophia optimizer, which also employs element-wise clipping as a backup update step.
> * Like Sophia, we found that controlled update magnitudes are crucial for stable training, especially in large-batch settings with higher learning rates.
> * Both optimizers demonstrate that properly designed clipping can enhance rather than harm optimization.
>
> This analysis provides deeper insights into how clipping contributes to optimization stability in large-batch training. Besides, we have included the visualization of the clipping ratio in GPT-2 Small training setting (2B tokens) for the 1st, 6th, and 12th layers in Appendix E:
>
> The clipping effect varies significantly across different layers:
> * Layer 1 (input layer) shows minimal clipping with ratios close to 0 throughout training, indicating that updates in early layers rarely require clipping.
> * Layer 6 (middle layer) shows moderate clipping ratios reaching up to 0.12 (12%) in later training stages.
> * Layer 12 (output layer) exhibits tiny clipping ratios (maximum ~0.0025 or 0.25%).
>
> This layer-wise analysis reveals that clipping primarily affects middle layers while having minimal impact on input and output layers. This pattern suggests that:
> * The clipping mechanism acts as a targeted stabilizer rather than a global constraint.
> * Most gradient updates proceed unmodified, preserving their original direction.
> * Middle layers, where feature representations are refined, benefit most from controlled updates.

---

> ### Author Response · Authors · 2024-11-20
> **Response to reviewer 1gMq (3/3)**
>
> **Q5: Analysis of "Curvature of Convergence Point" paragraph.**
>
> A5: Thank you for this insightful question about convergence properties. We can provide a more intuitive explanation of why MERIT tends to find flatter minima:
> MERIT's max-norm-based trust ratio directly constrains extreme weight values, particularly in attention layers.
> By controlling the max attention logit (as shown in Figure 2), MERIT prevents the optimization from entering regions with sharp curvature.
> The element-wise trust ratio calculation focuses on local weight structures, allowing for more balanced updates across different parts of the network.
>
> Because GPT-2 medium is larger and its whole hessian distribution cannot be directly accessed using PyHessian, instead we calculate the top eigenvalue and trace for each layer and present the mean value:
>
> |                 | AdamW | LAMB  | MERIT|
> |-----------------|--------|-------|--------|
> | Top Eigenvalue  | 114.643 | 19.7177 | 7.7236 |
> | Trace | 3766.64 | 1060.41 | 1051.45 |
>
> We find that MERIT consistently leads larger models to converge to flatter minima compared to LAMB and AdamW during large-batch training, as evidenced by the smaller top eigenvalue and trace of the Hessian matrix at the convergence point.
>
> **Q6: Figure 9 description.**
>
> R6: Figure 9 demonstrates validation loss results from large-batch training on the GPT2 small model, comparing different variants and learning rates. While MERIT achieved a validation loss of 3.280, all three ablation studies (removing Element-wise Clipping, Weight-wise Ratio Bound, or Element-wise Ratio) resulted in higher validation losses. We have revised the figure by adding the residual compared with the baseline.  This indicates that each of these mechanisms plays a crucial role in MERIT's effectiveness.

---

> ### Author Response · Authors · 2024-11-25
> **Looking forward to the reply**
>
> Dear reviewer 1gMq,
>
> Thank you for your time and valuable feedback on our work. We have carefully considered your comments and concerns, and have taken the following actions:
>
> 1. Clarified the current training limitations of larger language models and provided the experimental results of MERIT on the encoder-decoder (T5) model;
>
> 2. Analyzed the memory efficiency of MERIT in detail and revised minor typoes in the paper;
>
> 3. Provided a clear definition of clip function and included the visualization of the clipping ratio in GPT-2 small training with analysis;
>
> As we approach the end of the discussion period, please let us know if you have any remaining concerns or questions. We appreciate your time and efforts in helping us improve this work.

---

> ### Author Response · Authors · 2024-12-01
> **Additional Experimental Results on Llama**
>
> Dear Reviewer 1gMq,
>
> Thank you for your feedback. We have conducted additional experiments using the recently open-sourced LLaMA implementation (https://github.com/kyleliang919/C-Optim) to validate our findings further. Here are the corresponding results:
>
> |  | AdamW (lr=2e-3) | LAMB (lr=1e-2) | MERIT (lr=9e-3) |
> |--------|--------|------|-------|
> | Validation loss | 3.277 | 3.265 | 3.199 |
>
> We conducted our experiments according to the Chinchilla Law settings, training on 2.6 billion tokens from the C4 dataset using a batch size of 1K. We maintained the same hyperparameters as in our original paper, specifically a weight decay of 0.1 and beta values of 0.9 and 0.95. The results demonstrate that MERIT consistently improves performance across different language model architectures in large-batch training scenarios.
>
> As we near the end of the discussion period, we welcome any remaining questions or concerns you may have. We greatly appreciate your valuable input in improving this work.

---

> ### Comment · Reviewer_1gMq · 2024-12-03
>
> Thank you for your efforts in addressing the concerns and questions I raised in my review, and I apologize for the delay.
> I have now reviewed all the responses and feel that most of my concerns and questions have been adequately addressed. I would like to adjust my score accordingly.

---

> > ### Author Response · Authors · 2024-12-03
> > **Reply to Reviewer 1gMq**
> >
> > Thanks again for your thoughtful feedback. Your comments have significantly improved this work.

---

### Official Review · Reviewer_Xvrb · 2024-11-06

**Soundness:** 2
**Presentation:** 2
**Contribution:** 3
**Rating:** 5
**Confidence:** 3

**Summary:**

This paper discusses the optimization problem caused by too large maximum attention logits during large-batch training. The authors analyzed that the issue with LAMB arises from the use of L2-norm and proposed a new optimizer, MERIT, which utilizes the max norm to adjust extreme values and introduces element-wise trust ratios to enable more robust updates. In experiments based on GPT-2, MERIT is stable in large-batch settings and outperforms optimization methods such as AdamW and LAMB.

**Strengths:**

-	The problem of too large logist in attention is a problem that needs to be studied, and the motivation for this study is good.
-	A new optimization method is proposed, which is technically sound
-	In experiments with large batch sizes, MERIT outperformed several popular optimizers including AdamW and LAMB.

**Weaknesses:**

The writing of this paper needs considerable improvement.

-	The layout of figures and tables is confusing. For example, Figure 1 is shown on page 2 but is first mentioned on page 8.
-	Table 2 is shown on page 9 but is never mentioned in the content.
-	The caption description of the figure is confusing. It would be clearer to use \subfigure[]{}{} to display each subfigure separately and provide individual descriptions for each, rather than combining everything into one single caption for the entire figure.
-	 Some text descriptions are also unclear. For example, the caption in Figure 5 says "AdamW:3.470, LAMB:3.355..." The meaning of the numbers is confusing.
-	It is recommended to write the definition of Maximum normalized ratio in Section 4.1

There are also some experimental concerns.

-	What is the batch size of AdamW in the experiment? From Figure 7 and Table 2, we can see that the batch size of AdamW is 480 and that of MERIT is 4k/8k/12k. Why are they not set the same?
-	The ablation experiment in Figure 9 is a key part of verifying the contributions of this paper, but it only lists several absolute values under different learning rates without comparing the effects before and after ablation. I recommend that the authors redraw Figure 9 to include such comparisons.
-	The paper needs to compare the performance of MERIT under different batch sizes.

**Questions:**

-	In lines 46-50, the authors mention two challenges brought about by too large batch size. Can the method proposed in this paper handle these two challenges?
-	What is the connection between the issue of attention generating large logits and the large batch size? Why are these two problems not independent of each other? Could the authors elaborate on this further?

---

> ### Author Response · Authors · 2024-11-20
> **Response to reviewer Xvrb (1/2)**
>
> We sincerely thank the review Xvrb for the meticulous review and responsible attitude. We make responses as follows.
>
> **W1: The writing of this paper needs considerable improvement.**
>
> R1: Thanks for your advice on the layout of the paper, we have revised the paper accordingly for each mentioned point:
> 1. We have revised the introduction section to include the discussion of Figure 1 in lines 49-50.
> 2. We have included the discussion of results in Table 2 in line 429 to demonstrate the effectiveness of the proposed MERIT optimizer in the large-batch training of GPT-2 models.
> 3. To make the presentation of subfigures clearer, we add subtitles for each subfigure in the revised paper.
> 4. The numbers in the caption of Figure 5 mean the final converged validation loss of the corresponding optimizer for different sizes of GPT2 models.
> 5. In our original paper, we have included the definition of the Maximum Normalized Ratio in the caption of Figure 3 in section 4.1.
>
> **W2: What is the batch size of AdamW in the experiment?**
>
> R2: Thanks for your question. As illustrated in lines 346-348, for the main results, we set a batch size of 1K for GPT-2 small with 2B training tokens, 4K for GPT-2 medium with 8B tokens, and 8K for GPT-2 large with 16B tokens for the large-batch training setting for each optimizer for a fair comparison. In Figure 7 and Table 2, “AdamW-480-100K” means training the GPT2 small model using the batch size=480 with 100K steps, similarly, “LAMB-4k-12k” and “MERIT-4k-12k” mean training the GPT2 small model using the batch size=4k with 12K steps. We measure the comparison between AdamW small-batch trained GPT2 model and MERIT large-batch trained GPT2 model to demonstrate that MERIT enables larger batch size than LAMB with no performance gap (batch size=4k for small and batch size=6k for medium).
>
> **W3: The ablation experiment in Figure 9 is a key part of verifying the contributions of this paper, but it only lists several absolute values under different learning rates without comparing the effects before and after ablation.**
>
> R3: We appreciate your suggestion. Figure 9 shows the converged validation loss with the large-batch training setting on the GPT2 small model with different ablation variants and learning rate combinations. We have revised the figure by adding the residual compared with the baseline. Our MERIT optimizer achieves a validation loss of 3.280 (baseline) as shown in Figure 5, and we can find that three ablations (w/o Element-wise Clipping, w/o Weight-wise Ratio Bound, and w/o Element-wise Ratio) always present a higher validation loss than MERIT, which demonstrates the importance of three introduced mechanisms in our proposed algorithm.
>
> **W4: The paper needs to compare the performance of MERIT under different batch sizes.**
>
> R4: We appreciate this suggestion, but would like to clarify our research focus:
> * Research Scope:
> 1. Our work specifically targets large-batch training optimization.
> 2. Primary goal is enabling stable training at large batch sizes (4K-8K) in our chinchilla settings.
> 3. Not intended as a general-purpose optimizer for all batch sizes.
> * Key Evidence Already Provided:
> 1. Figure 1 shows better scaling behavior up to large batches than AdamW and LAMB.
> 2. Figure 7 demonstrates MERIT matches small-batch AdamW performance (batch=480) while using much larger batches:
>    - 4K batch for GPT-2 Small
>    - 6K batch for GPT-2 Medium
>
>    These results directly validate our optimizer's effectiveness in its intended use case
> * Focus on Practical Need:
> 1. The industry increasingly uses large batches to reduce training time.
> 2. Our experiments align with real-world large-batch training scenarios where GPT-style (decoder-only) languages are used.
> 3. Additional small-batch experiments would be tangential to our core contribution.
>
> Furthermore, we further test the performance using an 8K batch size to train GPT2-small model with Sophia setting (48K tokens) as an extension of Figure 7, the performance is shown below:
>
> | | AdamW | LAMB| MERIT |
> |-|--------|--------|--------|
> | Validation loss| 2.9873 | 2.9759 | 2.9433 |
>
> The results further confirm MERIT optimizer's superior performance in large-batch training.

---

> ### Author Response · Authors · 2024-11-20
> **Response to reviewer Xvrb (2/2)**
>
> **Q1: In lines 46-50, the authors mention two challenges brought about by too large batch size.**
>
> A1: We appreciate your question. The challenges in lines 46-50 focus on conventional explanations for why larger batch sizes lead to poorer generalization performance. Our research offers fresh perspectives on this issue by identifying a previously unrecognized connection: when training language models with large batches, there is a sharp increase in maximum attention logits, which hurts language models’ final generalization performance. To address this problem, we developed the MERIT optimizer by introducing a maximum-normalized element-wise ratio.
>
> **Q2: What is the connection between the issue of attention generating large logits and the large batch size?**
>
> A2: Thanks for your question. Because larger batch sizes reduce the total number of training iterations when a fixed number of training tokens is available, we need to use larger learning rates than small batch training. The proof of larger learning rates leading to larger max attention logits is provided as follows:
>
> $W_Q^{(t+1)}=W_Q^{(t)}−η∇_{W_Q}L$,
> $W_K^{(t+1)}=W_K^{(t)}−η∇_{W_K}L$
>
> $Q_i^{(t+1)}=X_iW_Q^{(t+1)}=X_i(W_Q^{(t)}−η∇_{W_Q}L) = X_iW_Q^{(t)} - ηX_i∇_{W_Q}L$, $K_j^{(t+1)}=X_jW_K^{(t+1)}=X_j(W_K^{(t)}−η∇_{W_K}L) = X_jW_K^{(t)} - ηX_j∇_{W_K}L$
>
> ${Logits}_{ij}^{(t+1)} = \frac{Q_i^{(t+1)} \cdot (K_j^{(t+1)})^\top}{\sqrt{d_k}}$
>
> $ = \frac{Q_i^{(t)} \cdot (K_j^{(t)})^\top}{\sqrt{d_k}} -\eta\frac{Q_i^{(t)}\cdot(X_j\nabla_{W_K}L)^\mathsf{T}}{\sqrt{d_k}}-\eta\frac{(X_i\nabla_{W_Q}L)\cdot K_j^{(t)\mathsf{T}}}{\sqrt{d_k}}+\eta^2\frac{(X_i\nabla_{W_Q}L)\cdot(X_j\nabla_{W_K}L)^\mathsf{T}}{\sqrt{d_k}}$
>
> Assuming that the gradients $∇_{W_Q}L$ and $∇_{W_K}L$ are not dependent on $η$ (which is typical in gradient descent), the change in the maximum logit is linearly proportional to $η$ because $\eta^2$ terms are negligible compared to $η$ terms:
>
> $\text{Max Logit}^{(t+1)} \approx \text{Max Logit}^{(t)} - η \cdot C + \mathcal{O}(η^2) \propto η $
>
> Our proof reveals that the maximum attention logit scales proportionally with $\eta$, which increases when using larger batch sizes during pre-training.

---

> ### Author Response · Authors · 2024-11-25
> **Looking forward to the reply**
>
> Dear reviewer Xvrb:
>
> Thanks so much again for the time and effort in our work. According to the comments and concerns, we conduct the corresponding experiments and further discuss the related points. Besides, we have revised our introduction section and several images accordingly. We also provided proof of the relationship between attention generating large logits and the large batch size.
>
> As the discussion period is nearing its end, please feel free to let us know if there are any other concerns. Thanks again for your time and efforts.

---

### Meta-Review · Area_Chair_KYad · 2024-12-14

**Metareview:**

The paper presents MERIT, a new optimizer for large-batch training of language models that mitigates max attention logit growth, which can harm performance. By using max norm for trust ratios and implementing element-wise trust ratios, MERIT enhances accuracy in capturing local weight structures.
Experiments on various GPT-2 models show that MERIT outperforms existing optimizers like AdamW and LAMB, allowing a batch size of 6,000 without performance loss, thus demonstrating its effectiveness in large-batch training.

Reviewers raised significant concerns about the paper's presentation, the fairness of comparisons with AdamW, and the insufficient empirical results for smaller models and a limited range of tasks. As it stands, the paper does not meet the standards required for ICLR.

**Additional Comments On Reviewer Discussion:**

Reviewers raised significant concerns about the paper's presentation, the fairness of comparisons with AdamW, and the insufficient empirical results for smaller models and a limited range of tasks. As it stands, the paper does not meet the standards required for ICLR.

---

### Decision · Program_Chairs · 2025-01-22

Reject